# GRAPHIX: A PRE-TRAINED GRAPH EDIT MODEL FOR AUTOMATED PROGRAM REPAIR

## ABSTRACT

We present GRAPHIX, a pre-trained graph edit model for automatically detecting and fixing bugs and code quality issues in Java programs. Unlike sequence-to-sequence models, GRAPHIX leverages the abstract syntax structure of code and represents the code using a multi-head graph encoder. Along with an autoregressive tree decoder, the model learns to perform graph edit actions for automated program repair. We devise a novel pre-training strategy for GRAPHIX, namely *deleted sub-tree reconstruction*, to enrich the model with implicit knowledge of program structures from unlabeled source code. The pre-training objective is made consistent with the bug fixing task to facilitate the downstream learning. We evaluate GRAPHIX on the *Patches in The Wild* Java benchmark, using both abstract and concrete code. Experimental results show that GRAPHIX significantly outperforms a wide range of baselines including CodeBERT and BART and is as competitive as state-of-the-art pre-trained Transformer models despite using fewer parameters. Further analysis demonstrates strong inductive biases of GRAPHIX in learning meaningful structural and semantic code patterns, both in abstract and concrete source code.

## 1 INTRODUCTION

Detecting bugs and code quality issues in programs and fixing them is an important task in software development. Oftentimes, this two-step process is manual and labor-intensive. In order to ease the burden on developers and reduce the development cost, several automated tools are developed and integrated with development environments (IDEs) and/or code review systems. There exist many program analysis techniques (Sadowski et al., 2015) that use hand-written rules to provide linter-style recommendations while others (Paletov et al., 2018) are able to detect specific bug types (e.g., crypto API uses) but leave them to the developer to fix. In the recent years, advances in deep learning have appealed researchers to move to data-driven approaches (Vasic et al., 2018) that learn to detect and fix bugs from data. Henceforth, we refer to the problem of automatically detecting and fixing bugs as *automated program repair* (Goues et al., 2019).

One main challenge with using machine learning for automated program repair is the lack of large human-labeled datasets. Several works (Tufano et al., 2019b;a) propose heuristics to extract bug-fix data from code changes. Code changes are represented as `(code before, code after)` pairs mostly at the method level, and typically pertain to feature enhancements, bug fixes and code refactoring. Code changes relating to bug fixes are identified through the presence of keywords such as "bug" in commit messages, through the size of changes or both. An advantage of using code change data is that there are plenty of code changes available in source code repositories.

Based on code change data, the learning of bug fixes has been taken in three main directions. The first line of work regards the bug-fix learning problem as sequence-based code generation and harnesses the power of sequence-to-sequence models with RNN (Hata et al., 2018; Chen et al., 2019; Tufano et al., 2019b;a) and more recently with pre-trained Transformer models (Feng et al., 2020; Guo et al., 2021; Ahmad et al., 2021; Wang et al., 2021; Berabi et al., 2021). Secondly, instead of generating the fixed code from scratch, Zhao et al. (2019); Chakraborty et al. (2020); Li et al. (2020); Panthaplackel et al. (2021) extend sequence models and propose to generate edits on the buggy code. Lastly, Yin et al. (2018); Tarlow et al. (2019); Dinella et al. (2020); Yao et al. (2021) deviate from the token-based representation of code and leverage the hierarchical structure in the abstract syntax tree

(AST). The high level idea is to learn a graph edit model to iteratively transform the AST of the buggy code into the AST of the fixed code. The learning is supervised by tree differencing edits via a cross-entropy loss. Despite the benefits of more precise bug localization, shorter tree edits and natural graph representations of programs, existing graph-based models have been less competitive than Transformer-based models thus far.

In this paper, we pursue the direction of representing explicit program structures with graphs and make contributions toward source code understanding using machine learning. Specifically, we present GRAPHIX, a pre-trained graph edit model for automated program repair. Inspired by HOPPITY (Dinella et al., 2020), we design GRAPHIX as a sequential decision process with a multi-head graph encoder and an autoregressive tree decoder. Conditioned on the graph state and the edit history at each step, the decoder iteratively makes primitive actions (e.g., adding a node) on the graph corresponding to the input AST. Unlike HOPPITY (Dinella et al., 2020), we enhance the encoder with multiple graph heads to capture diverse aspects of hierarchical code structures. In addition, we guide the decoder with an underlying Abstract Syntax Description Language (ASDL). Owing to the syntax language, the graph structure and semantic edits, GRAPHIX exhibits strong inductive biases in learning generic fixing and refactoring patterns from code changes. Moreover, we devise a novel pre-training strategy, namely *deleted sub-tree reconstruction*, that enables GRAPHIX to learn implicit program structures from unlabeled data. Here, the model is pre-trained to reconstruct a randomly deleted sub-tree given other code context, using the same cross-entropy loss as the program repair task. Our pre-training task generalizes the masked language model and denoising objectives (Devlin et al., 2019; Lewis et al., 2020) in sequence models to the AST representation of source code. Our idea shares the conceptual principle with generative models of code (Li et al., 2018; Brockschmidt et al., 2018) for expression generation and the structural language model (Alon et al., 2020) for any-code completion.

We evaluate GRAPHIX on both concrete and abstract versions of the *Patches in The Wild* Java benchmark (Tufano et al., 2019b). We show that GRAPHIX significantly outperforms edit-based and Transformer models, and the performance is as competitive as recent state-of-the-art pre-trained baselines despite using fewer parameters. Our in-depth analysis of the generated patterns demonstrates the ability of GRAPHIX to learn a wide range of meaningful and generic bug fixing and code refactoring patterns. Compared to HOPPITY, which merely learns short edits such as modifying access keywords (`private` to `public`), adding `break`, or removing redundant type arguments, etc., GRAPHIX is able to handle longer, more complicated structural and semantic edits (up to 20 graph edits). We showcase a variety of bug-fix examples such as fixing off-by-one errors, possible `null` pointer exceptions, etc. in our analysis.

In summary, we make the following main contributions in this work:

1. GRAPHIX is a medium-scale graph edit model that can be pre-trained with the *deleted sub-tree reconstruction* objective for automated program repair.

2. GRAPHIX significantly outperforms the edit-based models and is able to achieve high top-1 exact match accuracy on the *Patches in the Wild* benchmark even without pre-training. Using an explicit graph representation on the abstract syntax tree, our model is as competitive as large-scale Transformer models.

3. Unlike previous work, which only focused on the abstract version of the benchmark, we evaluate GRAPHIX on *both abstract and concrete code* and show the effectiveness of our model in both settings. This also suggests that the code abstraction may not be necessary.

4. Our in-depth analysis of the generated fixes demonstrates that GRAPHIX exhibits strong inductive biases in learning to both structural and semantic code patterns, leading many meaningful bug fixes.

Despite our focus on Java, our model and the pre-training strategy can be extended further to work with multiple languages for universal source code understanding using a language-agnostic abstract language and parser such as tree-sitter[1]. In addition, the syntax tree structure is highly amenable to program dependencies. We hope our work will inspire the community to advance the area further by incorporating more program dependencies, designing efficient graph learning algorithms with large model architectures and datasets.

---

[1]https://tree-sitter.github.io/tree-sitter/

## 2 RELATED WORK

**Deep learning for program repair** Advances in machine learning in the past decade have drastically changed the landscape of research for automated program repair. Inspired by Neural Machine Translation, Hata et al. (2018); Chen et al. (2019); Tufano et al. (2019b;a) regard code as a sequence of tokens and use sequence-to-sequence models to "translate" the buggy code into the fixed code. More recently, pre-trained Transformer-based models such as CodeBERT (Feng et al., 2020), GraphCodeBERT (Guo et al., 2021), PL-BART (Ahmad et al., 2021) and most recently CodeT5 (Wang et al., 2021) achieve state-of-the-art results on *"Patches in the Wild"* Java benchmark (Tufano et al., 2019b) for program repair. A series of edit-based approaches model the code edits made on the buggy code as opposed to the "translation"-based counterparts. Zhao et al. (2019); Chakraborty et al. (2020); Li et al. (2020); Panthaplackel et al. (2021) extend sequence-to-sequence RNN models to predict the sequence of edits given the buggy code. Unlike these works, we build GRAPHIX on the inherent abstract syntax structure of code instead of its token-based representation.

**Graph neural networks** GNNs such as graph convolutional networks (Kipf & Welling, 2016), graph attention networks (Veličković et al., 2018) provide a set of powerful tools for learning graph-structured data. Source code with rich graph structures in abstract syntax trees and program dependency graphs naturally fits to the GNN toolbox. Yin et al. (2018); Allamanis et al. (2018); Hellendoorn et al. (2019); Wang et al. (2020) augment the AST with additional data flow edges between variables and use GNNs for tasks such as variable name prediction. Yin et al. (2018); Tarlow et al. (2019); Dinella et al. (2020); Yasunaga & Liang (2020); Yao et al. (2021) use graph models based on ASTs for program repair and editing tasks in C, C# and Javascript, separately. Particularly, Brockschmidt et al. (2018); Dinella et al. (2020); Yao et al. (2021) use graph representations with tree-based decoders for AST generation by editing the partial AST one node at a time. Brockschmidt et al. (2018); Yao et al. (2021) further equip the decoder with an Abstract Syntax Description Language (ASDL) to ensure the syntactic correctness of the generated code. Our work advances HOPPITY (Dinella et al., 2020) and (Yao et al., 2021) with multi-head graph encoding and pre-training and demonstrates the benefits of program structures on learning bug-fix patterns in Java code.

**Pre-training** Transformer (Vaswani et al., 2017) and large-scale pre-training (Radford et al., 2018; Devlin et al., 2019; Lewis et al., 2020) have gained much traction in the broader field of natural language processing (NLP). Sequence-based pre-training with Transformer has also proven to be effective in learning source code representations. In contrast, techniques for pre-training with graphs are less well-developed, especially for learning graph representations of source code. Existing work on graph pre-training are mainly based on attribute and edge prediction (see (Hu et al., 2020) and references therein). Distinct from these works, our pre-training task is conceptually related to structural generative modeling of source code (Brockschmidt et al., 2018; Alon et al., 2020) for tree generation, just as pre-training techniques in BERT and BART derived from language modeling and denoising for sequence generation in NLP. However, we focus on pre-training GRAPHIX to learn hierarchical tree structures of code for program repair rather than on generating code.

## 3 MOTIVATING EXAMPLES

We begin by motivating the graph edit approach with two examples of code changes. Listing 1 displays two test samples from the *Patches in the Wild* benchmark, which are real-world Java code extracted from GitHub repositories.

In both examples, we can see that the code changes are much smaller compared to the method bodies. The fix of the first bug is straightforward with one edit that replaces `||` inside the condition with `&&` to prevent the program from crashing. For the second bug, the change is more structural and hierarchical than merely replacing the token `Object` with **void** and removing **return** keyword. Rather, the corresponding fix can be summarized by a sequence of four tree edits: (i) replace the return type `Object` with **void** type, (ii) remove the `ReturnStmt`, (iii) add an `ExpressionStmt` at the same position, (iv) copy the `AssignExpr` in the original code over the child of the newly added expression statement. More generally, each edit is associated with a specific node and consists of several primitive actions such as node deletion, addition or copy and type or value prediction.

These examples illustrate the benefits of the edit-based approach over the pure translation-based approach in three aspects. First, by modeling the edits, the model maintains the global code context

```java
public static List<String> getDomains(String consumerKey) throws APIManagementException {
    String list = ApiMgtDAO.getAuthorizedDomainsByConsumerKey(consumerKey);
    if ((list != null) || && (!(list.isEmpty()))) {
        return Arrays.asList(list.split(","));
    }
    return null;
}
```

```java
Object void setData(Object newData) {
    return data = newData;
}
```

Listing 1: Two bug-fix examples in the test *Patches in the Wild* benchmark Tufano et al. (2019b) GRAPHIX can detect and fix. In each method, we highlight added text in green and deleted text in red. We also remove package names from APIs and change the long name of the first method for readability. (top) A bug that may cause `NullPointerException` when `list = null` in the sample **medium/8390**. (bottom) A code quality issue detected in the sample **small/8616** where the setter has redundant `return` statement and type. The figure is best seen with color.

instead of decoding from scratch. Second, the edit sequence is often much shorter than the code sequence per se. Finally, with the structural edits, the decoding can be guided by an underlying syntax language. For example, the child of `ExpressionStmt` must be an expression (e.g., `AssignExpr` in the second bug fix) and must not be other production rules such as `ReturnStmt`. Another advantage is in terms of an end-to-end framework for bug detection and fixing in the sense that such an edit model can predict whether or not a program is buggy and localize the bug more precisely.

## 4 APPROACH

At a high level, we want to model $p(g_f|\theta, g_b)$ where $g_b$ is the buggy code and $g_f$ is the fixed code. Instead of encoding $g_b$ and generating $g_f$ from scratch, we learn a sequence of $T$ tree-edit actions $\{a_t\}_{t=1}^{T}$ that transforms $g_b$ to $g_f$ as motivated in Section 3. Indirectly, we learn the model $p(a_{1:T}|g_b)$ from a labeled dataset of samples, each with buggy code and the corresponding ground-truth edit, by minimizing a cross-entropy loss. Figure 1 illustrates the overall architecture.

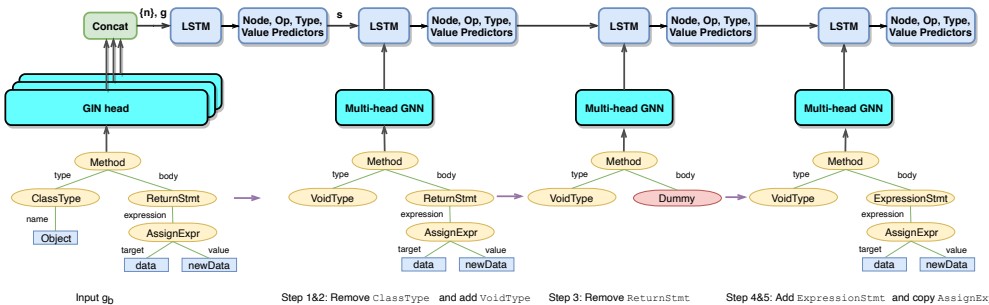

Figure 1: The overall architecture of GRAPHIX with an illustration of the edit sequence on the second buggy code in Listing 1. For readability, we show the partial AST with only two children of the root node. The graph encoder and decoder components are repeated for the illustration purpose.

One may recover the original token-based generation model when each action corresponds to adding a new token and $T$ is the length of $g_b$. The key departure from such a model is a more hierarchical program representation that allows shorter and more structural edits between the input $g_b$ and the output $g_f$. Inspired by (Dinella et al., 2020; Yao et al., 2021) which build on syntax structures and

tree differencing-based edits, we represent the buggy code as graph $g_b$ that is constructed from the program's syntax tree and extra data flow edges between certain leaf nodes.

Our model is sequential in nature with a multi-head graph encoder and an autoregressive tree decoder. At each time step $t$, the encoder encodes the current graph state $g_t$, and the decoder computes the graph edit history $s_t$. Conditioned on the current state, the decoder then predicts and performs tree edit actions. We discuss the baseline encoder, edit operations and ASDL-guided decoder in greater details in Appendix A. Next, we elaborate more on the multi-head graph encoder and our proposed pre-training strategy.

### 4.1 MULTI-HEAD GRAPH ENCODER

Earlier work  (Yin et al., 2018; Allamanis et al., 2018; Brockschmidt et al., 2018; Hellendoorn et al., 2019) has explored graph neural networks on rich graph structures of programs including syntax trees and dependency graphs for program representations. We follow (Dinella et al., 2020) and represent each program using its abstract syntax tree: the core graph is built upon the syntax tree with bidirectional edges between parent and child nodes, edges between adjacent leaf nodes and additional edges between value nodes to partially capture data flow information (Dinella et al., 2020; Allamanis et al., 2018). Then, we use a graph isomorphism network (GIN, Xu et al. (2019)) to compute a vector representation for each node and an aggregate graph representation. Motivated by the success of multi-head attentions in (Vaswani et al., 2017; Veličković et al., 2018), we propose a *multi-head graph encoder* for the graph encoding in which each head is a GIN and the head outputs are concatenated into corresponding node and graph embeddings. Using multiple graph heads allows the model to encode diverse aspects of the input program.

Our architecture is different from the *per-layer* multi-head attention in Transformer (Vaswani et al., 2017) and graph attention networks (Veličković et al., 2018) as well as multiple towers in message passing neural networks (Gilmer et al., 2017). We do not have cross-attention, hence having multiple heads per graph network layer is equivalent to adding more linear projections, which merely increase the GIN's depth. *We choose to concatenate the resulting GIN representations at the end as an ensemble model and find such a strategy effective in our ablation study* (c.f. Appendix E).

The graph encoder is the component with the most parameters. The encoder's size is the function of the number of layers, the latent dimension and the number of heads. We mainly experiment with a multi-head model that has 4 layers, 8 heads and the latent dimension $d = 256$, which amounts to 32M parameters.

### 4.2 PRE-TRAINING OBJECTIVE

We have described the architecture of GRAPHIX as a graph-based, autoregressive model. GRAPHIX is a supervised model and trained on labeled bug-fix data using the teacher-forcing technique. Since the labeled dataset is relatively small, in order for the model to learn more inherent program structures, we devise a novel strategy to pre-train GRAPHIX on unlabeled source code and allow the knowledge to be transferred to the downstream program repair task.

**Pre-training task.** Generalizing the masked language model pre-training idea in NLP (Devlin et al., 2019; Lewis et al., 2020), we devise a pre-training technique for the tree-structured data, namely *deleted sub-tree reconstruction*. To that end, we take the ASTs from a collection of unlabeled source code. For each AST, we randomly select a sub-tree of a certain size rooted at node $n$. We replace $n$ with a dummy node, and remove all the subsequent nodes and edges in the sub-tree. Given the dummy node's location, GRAPHIX is then trained to reconstruct the sub-tree using the other code context in the sequential manner. The decoder expands the partial AST one node at a time in the depth-first traversal order (Maddison & Tarlow, 2014; Bielik et al., 2016), using the production rules in the underlying abstract language. Similar to one tree edit in our bug-fixing task, each expansion step corresponds to a classification problem, and therefore we use the same cross-entropy objective for training without changing the network architecture.

Our pre-training task can be viewed as a generalization of the masked language model objective in BERT (Devlin et al., 2019) and a specialization of the denoising objective in BART (Lewis et al., 2020) to the tree structure of code. The graph reconstruction task is conceptually similar to the graph generative models (Li et al., 2018; Brockschmidt et al., 2018) for expression generation and the

structural language model of code (Alon et al., 2020) for any-code completion. Unlike these works, we are interested in pre-training rather than code generation/completion. Note that our pre-training technique is specific to ASTs with an underlying syntax language and not applicable to arbitrary graph structures. The pre-training strategy, however, can be extended to work with other types of graph corruptions such as multiple sub-tree additions and replacements as well as objectives such as link prediction and so on. We defer these directions to future work.

## 5 EXPERIMENTS

We evaluate GRAPHIX on the *Patches in the Wild* benchmark and compare its performance to existing translation-based and edit-based baselines. We conduct a series of experiments and studies on the network architectures (i.e., model size and single vs. multiple heads) and the data types (short vs. long function, and abstract vs. concrete code) to strengthen our evaluation.

### 5.1 DATASET

We use the *Patches in the Wild* Java bug-fix benchmark, which was curated and popularized by Tufano et al. (2019b). The dataset consists of 123,804 bug-fix pairs extracted from GitHub repositories. It has two subsets: the **small** subset contains 58,350 methods/functions with less than 50 tokens whereas each of the 65,454 **medium** functions is between 50 and 100 tokens. To build the final dataset, the authors extracted about 10M GitHub commits whose messages textually match one of these ("fix", "bug", "error", and "exception") patterns. Finally, the pair of code before and after each commit is considered one bug-fix sample with duplicates being removed. Each subset was originally divided into training, validation and test buckets, and we use the same splits in our evaluation.

**Code abstraction.** Tufano et al. (2019b) promoted the code abstraction idea where they abstracted away types, names and literal values in each method with generic names such as METHOD_1, VAR_2, INT_3. Most existing work employs this abstracted code benchmark with an only exception of (Drain et al., 2021). For convenience, we call it **abstract** benchmark. We also conduct extensive experiments on the **concrete** version that comes from the same dataset (Tufano et al., 2019b) without performing the code abstraction procedure.

### 5.2 DATA PROCESSING FOR GRAPHIX

Our proposed model GRAPHIX and its baseline HOPPITY (Dinella et al., 2020) operate on the AST level, so we need two additional processing steps: code parsing and ground-truth graph edit construction. Given each pair of buggy and fixed methods, we run JavaParser[2] to parse the source code into a pair of ASTs and serialize the output ASTs into a JSON format. We discarded 120 samples (out of 123,804 samples) that JavaParser was unable to parse. Those few samples should not affect the evaluation.

**Ground-truth edit construction.** Ground-truth edit sequences are the sole supervision signals that enable the learning of our graph edit model. Each edit includes a pre-defined operation type (see Appendix A.2 for more details), the edit location, the node type and value. HOPPITY (Dinella et al., 2020) uses JSON differencing algorithm[3] to create the ground-truth tree edit sequence for each pair of buggy and fixed code. Since GRAPHIX is a grammar-driven tree transformation (Yao et al., 2021), we implement a grammar-aware and shortest distance tree differencing algorithm, inspired by a dynamic program in Yao et al. (2021) (Algorithm 3). One could also use the GumTree differencing tool[4]. We show an example of two types of ground-truth edit sequences from the two differencing algorithms and statistics highlighting the advantages of the grammar-aware tree differencing in Appendix B.

**Pre-training data.** We use the CodeSearchNet dataset released by Husain et al. (2019) for our pre-training purpose. The entire dataset consists of 6.4M functions in six programming languages extracted from public non-fork GitHub repositories. Since our focus is on the *Patches in the Wild* Java benchmark, we use the Java portion of CodeSearchNet with a total of 1.5M functions. We filter out

---

[2]https://javaparser.org/
[3]2https://www.npmjs.com/package/fast-json-patch
[4]https://github.com/GumTreeDiff/gumtree

those functions that are not parsable or have more than 600 AST nodes. The remaining functions can be considered fixed code and are used the pre-training task. During pre-training, we select sub-trees between 2 and 6 descendants for the deletion and construct a sequence of addition operations as the ground-truth edits. More details on implementation are provided in Appendix C.

### 5.3 BASELINES AND METRICS

We compare GRAPHIX to translation-based models including an LTSTM model (Tufano et al., 2019b) as well as state-of-the-art pre-trained Transformer models including notably PL-BART (Ahmad et al., 2021) and CodeT5 (Wang et al., 2021). We also compare our model against edit-based models including *Copy That!* (Panthaplackel et al., 2021) and HOPPITY (Dinella et al., 2020) of which the latter is directly related to our work. HOPPITY applies to Javascript and only works on short single-node edits (up to 3). That is, it does not handle a non-trivial sub-tree addition. For comparison, we modify its implementation to handle Java source code, and in order to add a sub-tree, we flatten the sub-tree from its root in the top-down and left-right order and create a series of single-node addition edits. To train the model, we use the teacher forcing technique so that at each intermediate step of the addition, the model is given the location of the parent node as well as type and value in the previous step. We name this enhanced baseline e-HOPPITY for convenience.

**Metrics.** We use the standard top-1 exact match accuracy (EM) as the metric and compare the performance of all the above models on the **small** and **medium** test sets. Note that EMs for e-HOPPITY and GRAPHIX are computed based on AST matching, which is equivalent to sequence matching in the other models. For GRAPHIX and e-HOPPITY, we use beam search of size 5 during inference and select the candidate with the highest score normalized by the edit sequence length.

### 5.4 RESULTS

In this section, GRAPHIX refers to the 8-head model with 32M parameters that is not pre-trained. GRAPHIX-P refers to GRAPHIX that is pre-trained and then fine-tuned on the respective datasets. We present the results on the **abstract** and **concrete** benchmark in Table 1 and Table 2, respectively. Each table consists of three groups of results: the first group represents the translation-based methods without pre-training, the second group is for edit-based methods and the last group includes pre-trained models including GRAPHIX-P. Note that DeepDebug (Drain et al., 2021) was not evaluated on the **abstract** code while most were not on the **concrete** code.

Table 1: Top-1 exact match (EM) accuracy on the **abstract** benchmark **with** and **without** pre-training.

| Model | Pre-trained Data | Size | Small | Medium |
|---|---|---|---|---|
| LSTM (Tufano et al., 2019b) | | 10M | 9.22% | 3.22% |
| GRU + Token Copy (Panthaplackel et al., 2021) | None | 250K | 14.80% | 7.00% |
| Transformer (Drain et al., 2021) | | 60M | 11.10% | 2.70% |
| GRU + Span Copy (Panthaplackel et al., 2021) | | 250K | 17.70% | 8.00% |
| e-HOPPITY (Dinella et al., 2020) | None | 1M | 7.30% | 1.21% |
| GRAPHIX | | 32M | **18.20**% | **9.19**% |
| CodeBERT (Feng et al., 2020), | CodeSearchNet | 180M | 16.40% | 5.20% |
| GraphCodeBERT (Guo et al., 2021) | CodeSearchNet | 110M | 17.30% | 9.10% |
| BART (Drain et al., 2021) | Large (54GB) | 400M | 16.70% | 6.70% |
| PL-BART (Ahmad et al., 2021) | Large (655GB) | 140M | 19.21% | 8.98% |
| CodeT5-small (Wang et al., 2021) | CodeSearchNet+ | 60M | 19.06% | 10.92% |
| CodeT5-base (Wang et al., 2021) | CodeSearchNet+ | 220M | **21.61**% | **13.96**% |
| GRAPHIX-P | CodeSearchNet Java | 32M | 19.81% | 8.81% |

From Table 1, we can see that our models significantly outperform the other edit-based baselines and are on a par with larger pre-trained models. Specifically, without pre-training GRAPHIX improves CodeBERT by 1.8% on the **small** subset and 4% on the **medium** subset. We observe a similar improvement of GRAPHIX over BART. When pre-trained on CodeSearchNet Java, GRAPHIX-P slightly outperforms PL-BART, CodeT5-small on the **small** but underperforms them on the **medium**. CodeT5-base with 220M parameters is currently the state-of-the-art performance on both datasets.

Table 2: Top-1 exact match (EM) accuracy on the **concrete** benchmark **with** and **without** pre-training.

| Model | Pre-trained Data | Size | Small | Medium |
|---|---|---|---|---|
| Transformer (Drain et al., 2021) | | 60M | 14.60% | 3.70% |
| GRU + Token Copy (Panthaplackel et al., 2021) | None | 10M | 6.80% | N/A |
| DeepDebug (T5) (Drain et al., 2021) | | 400M | 13.90% | 3.60% |
| GRU + Span Copy (Panthaplackel et al., 2021) | | 250K | 9.20% | N/A |
| e-HOPPITY (Dinella et al., 2020) | None | 1M | 7.28% | 1.51% |
| GRAPHIX | | 32M | **17.87%** | **9.01%** |
| DeepDebug (T5) (Drain et al., 2021) | Java (54GB) | 400M | 16.80% | 6.30% |
| DeepDebug (T5) (Drain et al., 2021) | English and Java | 400M | 18.70% | **11.40%** |
| PL-BART (See below) | Large (655GB) | 140M | **19.81%** | 6.37% |
| GRAPHIX-P | CodeSearchNet Java | 32M | 19.31% | 8.69% |

Table 2 shows the results of the models on the **concrete** benchmark. The PL-BART result, which was not reported in the original paper, is obtained by fine-tuning the released pre-trained PL-BART model on the concrete code. We observe a similar improvement on the **concrete** benchmark for GRAPHIX-P over GRAPHIX, which is significantly better the edit baselines, LSTM and the Transformer model. GRAPHIX-P is slightly better than DeepDebug (Drain et al., 2021) on the **small** set but worse on the **medium**. Compared to PL-BART, GRAPHIX-P performs comparably on the **small** dataset and outperforms PL-BART by a good margin on the **medium** set. In short, our model works comparably to DeepDebug and PL-BART on the concrete benchmark despite much fewer parameters.

Comparing the results in Table 1 and Table 2, we see a negligible drop about 0.2 – 0.5% in accuracy of GRAPHIX from the abstract to concrete code. On the contrary, the accuracy of GRU with span copy (Panthaplackel et al., 2021) decreases by almost half. This demonstrates that GRAPHIX is insensitive to the naming of variables, types and APIs. We also observe a similar trend for PL-BART. These observations suggest that the code abstraction may not be necessary. Additionally, we notice that while the pre-training provides an additional 10% relative gain on the **small** subsets, GRAPHIX-P does not achieve similar gains on the **medium** set, which is inherently longer and harder. We hypothesize that the synthetic edits used for pre-training might be more aligned with the **small** dataset than the **medium**, and more sampling strategies may be needed to bridge the gap.

## 6 ANECDOTAL EXAMPLES

In addition to several state-of-the-art results in EM accuracy, GRAPHIX demonstrates strong inductive biases in learning complex bug-fix patterns. In this section, we showcase several bug examples that GRAPHIX is able to detect and fix. We show more such examples in Appendix F.

```java
public void stopLocationUpdates() throws SecurityException {
    if ((locationManager) != null) {
        locationManager.removeUpdates(this);
    }
}
```

```java
public static Throwable getRootCause(Throwable t) {
    if (t == null)           return null;
    Throwable rootCause = t;
    Throwable cause = rootCause.getCause();
    while ((cause != null) && (cause != rootCause)) {
        rootCause = cause;
        cause = cause.getCause();
    }
    return rootCause;
}
```

Listing 2: Missing `null` check bugs in the samples **small/11219** (top) and **medium/10509** (bottom).

```java
@Override
public void resolveAnaphora() {
    List<Proposition> props = VariableStorage.getPopostionList();
    int i = 0;
    for (CQuantifer quant : this.getQuantifers()) {
        if (i < = (props.size()))
            props.get(i).setLinkedId(((String) (quant.getVar().getSourceId()))));
        i++;
    }
}
```

```java
public static int mul(int n1, int n2) {
    return n1 + * n2;
}
```

Listing 3: (top) Off-by-one error in sample **medium/11874** (including = may cause an index-out-of-bound exception) and (bottom) logical bug in sample **small/8295**.

We manually analyze the generated fixes of GRAPHIX on the test **concrete** benchmark. Since longer edits tend to have lower probability, we normalize the prediction score of each generated fix by the length of the corresponding generated edit sequence. More specifically, we divide the negative log-likelihood by the edit length and exponentiate the result to obtain the normalized score. Then, we manually check 500 correct and incorrect fixes with highest scores in each of the small and medium sets. By inspecting the generated edits, we can easily spot out the code changes in each case. We find that similar fixes have similar edits and scores. This analysis enables us to recognize interesting bug-fix patterns as well as noises in the data. The patterns range from syntactic fixes such as using `equals()` instead of `==` or simplifying logic statements to semantic ones such as like adding **null** check conditions, fixing variable misuses and so on. Here, we show some exemplary samples of fix patches that match the ground-truth in the benchmark. Listing 2 displays two examples of potential bugs where the developers fail to validate the input/a class field. GRAPHIX can not only detect missing input validations (i.e., checking **null**), it can also return an intended behavior in the second example when the input is **null**). Listing 3 shows two other examples where GRAPHIX fixes an off-by-one error and a logical bug by apparently understanding the method name `mul`.

We show several instances of incorrect fixes generated by GRAPHIX in Appendix G. We note that extra enclosed parentheses `()` in the examples are artifacts of our parser. Finally, GRAPHIX and GRAPHIX-P share similar inductive biases and hence share similar sets of bug-fix patterns.

# 7  DISCUSSION AND FUTURE WORK

In this paper, we consider the problem of generating bug fixes using a neural model. We present GRAPHIX, a multi-head graph edit model that is pre-trained based on deleted sub-tree reconstruction. GRAPHIX establishes the high performance on a Java benchmark. Anecdotal fixes demonstrate that the model is able to detect and suggest fixes for a diverse set of bugs and program issues. In particular, GRAPHIX is about $2\times$ smaller than state-of-the-art Transformer models. Finally, GRAPHIX performs equally well on generating abstract as well as concrete fixes.

**Practical use.** The problem of bug detection is nuanced, and several times ML models can be replaced by simple rules. We have shown that GRAPHIX can capture semantic bug fixes through dozens of anecdotal examples (see Appendix F for more). In addition, we demonstrate in Appendix D that one can calibrate the model score to achieve an increased precision up to 57% by trading-off the coverage of bug detection.

We anticipate several directions in the future work: (i) pre-train based on multiple sub-tree additions and replacements (ii) incorporate more program dependencies into the graph representation and (iii) leverage Transformer and GNNs in a unified architecture for the global code context in the sequence.

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

# Appendix

## A ENCODER AND DECODER

This section describes the technical formulation of the single-head encoder and the auto-regressive decoder. For more details into these two components, we refer the reader to (Dinella et al., 2020; Yao et al., 2021).

### A.1 SINGLE-HEAD ENCODER

Formally, given a graph $g = (V, E)$ with a set of nodes $V$ and edges $E$, a single-head graph encoder gives a $d$-dimensional representation of the graph $g$ and representations of individual nodes $v \in V$. Similar to (Dinella et al., 2020), we compute the graph presentations for each edge type (i.e., AST edges, last-variable-use edges and successive edges) as follows:

$$h_v^{(l+1,k)} = \sigma \left( \sum_{u \in N^k(v)} W_1^{l,k} h_u^{(l)} \right), \forall k \in \{1, 2, \ldots, K\} \tag{1}$$

$$h_v^{(l+1)} = \sigma \left( W_2^l \left[ h_v^{(l+1,1)}, h_v^{(l+1,2)}, \ldots, h_v^{(l+1,K)} \right] + h_v^{(l)} \right), \tag{2}$$

where $K$ denotes the total number of edge types, $L$ is the number of propagations. $W_1^{l,k} \in \mathbb{R}^{d \times d}$ and $W_2^{l,k} \in \mathbb{R}^{dK \times d}$ are the network weights. $N^k(v)$ denotes a set of neighbors of $v$ that are connected to $v$ with the edge type $k$. These formulae give the node embeddings $\boldsymbol{n} = \{h_v^{(L)}\}_{v \in V}$ for all nodes $v \in V$ while the graph representation is an aggregation of $h_v^{(l)}$ for all $l$. More precisely, we use max pooling to combine $h_v^{(l)}$ for each $l$ and average the $L + 1$ resulting vectors to compute the final graph representation $\boldsymbol{g}$.

The initial embedding $h_v^{(0)}$ vector of node $v$ is set as the sum of the embedding of its node type and node value if the value exists. The node type and value embeddings are stored in separate tables.

### A.2 TREE EDIT OPERATIONS

GRAPHIX is a sequential decision process that iteratively performs tree edits on the partial tree at each step. Following (Brockschmidt et al., 2018; Yao et al., 2021), we employ a grammar-based tree decoder to ensure the grammatical correctness of each edit action. To that end, we construct an abstract language based on JavaParser[5]. The grammar includes a set of primitive and composite types (c.f. (Rabinovich et al., 2017; Yin & Neubig, 2017)). Each composite type (e.g., `stmt`) has a collection of production rules such as `ExpressionStmt(expr expression)` where each rule is represented as a constructor with a list of field arguments. Each argument has the field type (e.g., `expr`), the field name (e.g., `expression`) and the cardinality of the field as *single*, *optional*, or *sequential*. The cardinality indicates the number of children a node accepts as grammatically valid children. Based on the grammar, we define the following primitive operations:

`DELETE` operator takes a tree node $n$ and removes it from the tree. As a result, the corresponding sub-tree and the node $n$ are removed. However, removing a node that corresponds to a *single* field results in a grammatically invalid tree. For instance, removing node `ClassType` in step 1 of Figure 1 causes the AST to have no type, which is invalid. To ensure the syntax validity at each edit step, after the removal, we add a `Dummy` node as a placeholder with no node type or value. Another benefit of `Dummy` node is to reduce the uncertainty of the node selection in the subsequent step.

`ADD` operator adds a node to the graph. Under the grammar and the dummy node mechanism, the addition of a new node occurs in two cases. In the first case, we add a new child to a parent node of a *sequential* field. For example, the node of type `(stmt* statements)` may have an arbitrary number of children. In this case, adding a node requires the location of the parent node and the position to be added in its children list. In the second case, we add a new node into the position of a `Dummy` node (adding a *non-terminal* node `ExpressionStmt` in Figure 1 in step 4). Similarly to

---

[5]https://javaparser.org/

the `DELETE` operation, once adding a non-terminal node, we follow the underlying grammar specified by the added production rule and instantiate the set of dummy, child nodes at the newly added node. The `ADD` operator is also used to populate empty *terminal* nodes with actual node types and possible values. Note that while constructing the original tree, we also add `Dummy` at the empty fields.

`COPY` operator is introduced to simplify and shorten the edit sequence. While making code changes, the developer often moves code one place to another. Such changes can be translated directly on the AST by copying the corresponding sub-tree from the input AST. In essence, this operator locates the target position similarly to the `ADD` operation and then copies the sub-tree from the original tree to the target position of the current tree in one single step. For example, the subtree for the assignment expression `data = newData` in Figure 1 at $t = 2$ is copied to the new position in step 6.

Finally, the `NO_OP` action is used to automatically terminate the iterative tree editing procedure. After this step, the remaining dummy nodes will be pruned.

### A.3 AUTOREGRESSIVE TREE DECODER

Given the graph representation $\boldsymbol{g}_t$ of the program at step $t$ and the edit history $\boldsymbol{s}_{t-1}$ from the previous step, we use an LSTM-based tree decoder to update the state $\boldsymbol{s}_t = \text{LSTM}(\boldsymbol{g}_t|\boldsymbol{s}_{t-1})$. Based on the updated state $\boldsymbol{s}_t$, the model predict edit actions where each node edit includes a node selector, an operator predictor and a type/value predictor. The node selector selects node $loc_t$, then the operator predictor decides which operator $op_t$ to apply the edit on $loc_t$ among the four possible operators $\{\text{DELETE}, \text{ADD}, \text{COPY}, \text{NO\_OP}\}$. For the `ADD` operation, the type predictor further determines the node type/label $type_t$ for the newly added node. For terminal nodes, the value predictor computes the node value $value_t$. The probability of an edit is then decomposed as

$$p(a_t|\boldsymbol{s}_t) = p(loc_t|\boldsymbol{s}_t)p(op_t|\boldsymbol{s}_t, loc_t)p(type_t|\boldsymbol{s}_t, op_t, loc_t)p(value_t|\boldsymbol{s}_t, op_t, loc_t, type_t). \quad (3)$$

We provide more details into one decoding step below:

**Node selection.** For node selection, we employ a pointer network (Vinyals et al., 2015) since different graphs have different number of nodes. For `ADD`, the selector either selects a sequential-field node or a `Dummy` node. In the first case, it determines the parent of the node to be added and its left sibling. Therefore, we further decomposte $p(loc_t|\boldsymbol{s}_t) = p(sibling_t|\boldsymbol{s}_t, parent_t)p(loc_t|\boldsymbol{s}_t, parent_t)$. If $parent_t$ does not have any children, the left sibling node is considered as $parent_t$ itself. We compute the logits by taking inner product between the edit history $\boldsymbol{s}_t$ and the node embeddings $\boldsymbol{n}_t$. We select the node that maximizes the inner product. After selecting the node, we update the state with an additional LSTM call $\boldsymbol{s}_t = \text{LSTM}(\text{emb}(loc_t)|\boldsymbol{s}_t)$ where $\text{emb}(loc_t)$ is the node embedding of the selected node $loc_t$.

**Operator prediction:** The operator prediction is a regular classification problem with four labels $\{\text{DELETE}, \text{ADD}, \text{COPY}, \text{NO\_OP}\}$. We calculate the probability of operator $op_t$ using a softmax layer such that $p(op_t|\boldsymbol{s}_t, loc_t) = \text{softmax}(\boldsymbol{W}_{op}\boldsymbol{s}_t + \boldsymbol{b}_{op})$. After that, we again update the state via $\boldsymbol{s}_t = \text{LSTM}(\text{emb}(op_t)|\boldsymbol{s}_t)$ where $\text{emb}(op_t)$ is the embeding of the selected operator $op_t$.

**Type prediction.** This module assigns a type $type_t$ to the newly added node in the AST. Since the number of types is specified *a priori* by the underlying syntax language, we simply treat the type predictor as a classification problem over all possible types while masking out the impossible choices. We calculate the probability of operator $op_t$ using a softmax layer such that $p(type_t|\boldsymbol{s}_t, loc_t) = \text{softmax}(\boldsymbol{W}_{type}\boldsymbol{s}_t + \boldsymbol{b}_{type})$. We update the state via $\boldsymbol{s}_t = \text{LSTM}(\text{emb}(type_t)|\boldsymbol{s}_t)$ where $\text{emb}(type_t)$ is the embedding of the newly predicted type $type_t$.

**Value Prediction.** If the newly added node is a terminal node, the value predictor assigns to it a value $value_t$. The possible values are chosen from the union of the local values present in the input (i.e., local value table) and the global values that frequently appear in other programs. The local value embeddings are computed by the graph encoder while the global value embeddings are stored globally in the value embedding table. We compute the scores by simply taking inner product between the state $\boldsymbol{s}_t$ and the value embeddings and select the one maximizing the score. Finally, we update the state via $\boldsymbol{s}_t = \text{LSTM}(\text{emb}(value_t)|\boldsymbol{s}_t)$ where $\text{emb}(value_t)$ is the embedding of the value $value_t$.

The decoding process stops once it encounters `NO_OP` or reaches the maximum number edits.

## B  COMPARISON OF TREE DIFFERENCING ALGORITHMS

Table 3 shows the advantages of structural, grammar-aware tree differencing with copy operations in GRAPHIX over JSON differencing used in HOPPITY.

Table 3: Comparison of the edit sequence length between JSON differencing and grammar-aware differencing algorithms.

| Differencing | Type | No. of tokens | Mean | Median | 75th quantile |
|---|---|---|---|---|---|
| JSON differencing | Small | $< 50$ | 14.1 | 6 | 20 |
|  | Medium | 50-100 | 35.2 | 14 | 41 |
|  | Combined | $\leq 100$ | 25.2 | 9 | 31 |
| Grammar-aware differencing | Small | $< 50$ | 5.4 | 2 | 7 |
|  | Medium | 50-100 | 8.5 | 4 | 10 |
|  | Combined | $\leq 100$ | 7.1 | 3 | 9 |

**medium/6715**

```java
@Override
public boolean create(POJO.User user) throws SQLException {
    stmt = connect.prepareStatement("INSERT INTO User ...");
    stmt.setString(1, user.getPseudo());
    stmt.setString(2, user.getPassword());
    stmt.setString(3, user.getEmail());
    stmt.executeUpdate();
    stmt.close();
    System.out.println(...);
    return true;
}
```

**JSON differencing edits**

```
1. Remove the statement in line 10: return true;
2. Remove the scope: System.out
3. Remove the arguments in println
3. Remove the name println
4. Replace the remaining MethodCallExpr with BooleanLiteralExpr
5. Add true
6. Replace ExpressionStmt with ReturnStmt
```

**Grammar-aware differencing edits**

```
1. Remove the statement in line 9: System.out.println(...);
```

Listing 4: An example of code change where the `println` statement in line 9 is removed. The structural edit is straightforward with one remove step. The JSON differencing edits are less natural: the algorithm first removes line 10 and gradually changes line 9 to re-add the **return true** statement.

We can see that the structural edit differencing results in much shorter edit sequences on average. Examination of edits shows that structural edits also more meaningful. We show an example of code change and edits in Listing 4.

## C  IMPLEMENTATION AND TRAINING DETAILS

We implement GRAPHIX with PyTorch based on the open source implementation of HOPPITY (Dinella et al., 2020). We train GRAPHIX on a machine with 8 V100 GPUs, each with 32GB memory. We use batch size 5 on each GPU both in pre-training and fine-tuning. We use the Adam optimizer

and a linear learning rate scheduler, used in Transformer (Vaswani et al., 2017). We set the initial learning rate $10^{-4}$ and the warmup steps corresponding to 1 epoch. We pre-train GRAPHIX for 10 epochs for 10 days. We fine-tune GRAPHIX for 20 epochs with the maximum number of edit steps being 20. The fine-tuning takes about 2 hours per epoch on the small subset and about 3.5 hours per epoch on the medium set. We choose the best model checkpoint using the validation set and report its top-1 accuracy on the test set.

## D    HIGH PRECISION REGIME FOR GRAPHIX

As shown previously and in Appendix F, GRAPHIX is able to learn meaningful and interpretable code patterns, leading to a wide range of interesting bug fixes. Indeed, the actual kinds of fixes have not been studied thoroughly for the pre-trained Transformer models. Moreover, we show that our model can be of potential use in practice by trading-off the recall for an increased precision. To demonstrate that, we show the impact of calibrating the prediction scores to achieve such a trade-off.

Table 4: A trade-off between the accuracy and bug detection coverage.

| Theshold | 0.5 | 0.6 | 0.7 | 0.8 | 0.9 |
|---|---|---|---|---|---|
| Accuracy | 29.27% | 38.12% | 47.37% | 56.59% | 52.38% |
| No. predictions | 1882 | 926 | 418 | 129 | 21 |

Table 4 suggests that we can achieve higher accuracy at the cost of lower prediction frequency. For example, the model can suggest 418 most probable bug fixes at 47.37% correct patches, at the threshold 0.7 on the prediction score.

## E    ABLATION STUDIES

### E.1    EFFECTS OF THE GRAPH MODEL SIZE AND PRE-TRAINING

In Table 5, we compare GRAPHIX with HOPPITY using about 1M parameters. Here, GRAPHIX-B refers to the proposed GRAPHIX model with 1M parameters. The results show that GRAPHIX is significantly better than HOPPITY because of the ASDL-guided decoding and the grammar-aware shortest edit differencing. To be specific, GRAPHIX edits are significantly shorter and more meaningful than JSON differencing edits (see Table 3 and Listing 4 in Appendix B for the comparison). Moreover, the grammar-constrained decoder reduces the uncertainty of predicting edit properties at each decoding step.

Table 5:  Effects of the graph model size and pre-training on GRAPHIX.

| Setting | Model | Size | Abstract | | Concrete | |
|---|---|---|---|---|---|---|
| | | | Small | Medium | Small | Medium |
| No pre-training | e-HOPPITY | 1M | 7.30% | 1.21% | 7.28% | 1.51% |
| No pre-training | GRAPHIX-B | 1M | 12.40% | 5.34% | 11.21% | 4.31% |
| No pre-training | GRAPHIX | 32M | 18.20% | **9.19%** | 17.87% | **9.01%** |
| With pre-training | GRAPHIX-P | 32M | **19.81%** | 8.81% | **19.31%** | 8.69% |

We also study the effects of the graph model size and pre-training on GRAPHIX. Table 5 suggests that the multi-head encoding leads to about 50% relative boost in accuracy on the **small, abstract** subset and almost $2\times$ boost on the **abstract, medium**. The improvements are consistent on the concrete code. It can be seen that the pre-training gives an additional 10% relative gain on the small subsets but does not help on the medium sets. We explore more pre-training options and model sizes in our future work.

Next, we study the role of our proposed multi-head architecture.

### E.2 ROLE OF THE MULTI-HEAD ARCHITECTURE

Table 6: Role of the multi-head architecture when the dimension $d$ and the number of heads $H$ vary.

| Model | $H$ | $d$ | Size | Abstract | | Concrete | |
|---|---|---|---|---|---|---|---|
| | | | | Small | Medium | Small | Medium |
| GRAPHIX-B | 1 | 128 | 1M | 12.40% | 5.34% | 11.21% | 4.31% |
| | 1 | 256 | 4M | 14.94% | 7.07% | 16.58% | 7.79% |
| | 4 | 128 | 4M | 15.37% | 7.88% | 16.79% | 8.34% |
| | 1 | 566 | 32M | 14.63% | 6.52% | 15.95% | 6.63% |
| GRAPHIX | 8 | 256 | 32M | **18.20%** | **9.19%** | **17.87%** | **9.01%** |

Following (Dinella et al., 2020), we use the GIN depth of 4 for all the experiments. To understand the the impact of the number of heads compared to the latent dimension, we experiment with the latent dimension $d$ and the number of heads $H$ such that the corresponding model sizes are comparable. It can be seen from Table 6 that the multi-head models outperform the corresponding single-head counterparts with comparable sizes, for both 4M and 32M models. Moreover, increasing the model capacity with $d$ improves the performance as $d$ increases from 128 to 256 while the model with $d = 566$ tends to overfit the training data and performs poorly on the test set. In summary, our experiments show the benefits of the proposed multi-head architecture.

### E.3 PER-LAYER VERSUS LAST-LAYER MULTI-HEAD

We implement a per-layer multi-head architecture with head averaging and GIN, following the idea in GAT (Veličković et al., 2018). In this architecture, each head computes a linear projection of the input before it is fed to the respective GIN layer, so without attention having multiple heads per layer is the same as adding two more consecutive linear layers before that GIN layer: the first layer stacks the linear projection matrices into one weight matrix whereas the second has a fixed weight and computes average. We take the average of the head outputs instead of the concatenation for a computational reason.

We compare this new model with a single-head model of the same latent dimension ($d = 256$), and our ensemble-style multi-head models (with $H = 4$ and $H = 8$ heads). The results and configuration of each model are shown in Table 7.

Table 7: Effects of the per-layer versus last-layer multi-head architectures.

| Head type | $H$ | Agg. | $d$ | Size | Abstract | | Concrete | |
|---|---|---|---|---|---|---|---|---|
| | | | | | Small | Medium | Small | Medium |
| Single-head | 1 | N/A | 256 | 4M | 14.94% | 7.07% | 16.58% | 7.79% |
| Last-layer MH | 4 | Concat | 128 | 4M | 15.37% | 7.88% | 16.79% | 8.34% |
| Per-layer MH | 8 | Average | 256 | 6M | 15.78% | 7.40% | 16.18% | 7.93% |
| Last-layer MH (GRAPHIX) | 8 | Concat | 256 | 32M | 18.20% | 9.19% | 17.87% | 9.01% |

We can see this model performs slightly better than the single-head model thanks to the additional multi-head layers across the datasets, but it is worse than GRAPHIX with 4 heads at the end and 2M fewer parameters, except for the **abstract, small** dataset. Along with the previous empirical study on the width, this experiment demonstrates the benefits of the proposed multi-head architecture over purely increasing the graph neural network's depth or width.

## F MORE ANECDOTAL EXAMPLES

We show some more bug fixing and code refactoring patterns learned by GRAPHIX. In some cases, we show the pairs of buggy code and fixed code. In other cases, we only highlight the changes with added text in green and deleted text in red. A few long statements are replaced with ".. .".

### F.1 SIMPLIFYING LOGIC EXPRESSIONS

| Before | After |
|---|---|

**small/6886**

```java
public boolean isEmpty() {          public boolean isEmpty() {
    if ((first) == null) {              return (first) == null;
        return true;                }
    }
    return false;
}
```

**medium/7382**

```java
@Override
public boolean onTouchEvent(android.view.MotionEvent ev) {
    super.onTouchEvent(ev);
    dragHelper.processTouchEvent(ev);
    ViewGroup parent = ((ViewGroup) (findBottomView(this, x, y).getParent()));
    return false || (parent == (this));
}
```

### F.2 FIXING OFF-BY-ONE ERRORS

| Before | After |
|---|---|

**small/6229**

```java
@Override                           @Override
public boolean hasNext() {          public boolean hasNext() {
    return ((cursor) + 1) < (batches);   return (cursor) < (batches);
}                                   }
```

**small/9321**

```java
public boolean hasNext() {          public boolean hasNext() {
    return (frameIndex) < ((count) - 1);   return (frameIndex) < (count);
}
```

**medium/11106**

```java
public static boolean isPowerOfTwo(long number) {
    if (number <= 0) {
        throw new IllegalArgumentException(...);
    }
    if ((number & (-number)) == number) {
        return true;
    }
    return false;
}
```

**medium/10998**

```
public int run() {
    int exponent = 1000;
    BigInteger base = BigInteger.valueOf(2);
    BigInteger value = BigInteger.ZERO;
    int sum = 0;
    value = base.pow(exponent);
    String str = value.toString();
    for (int i = 0; i < ((str.length()) - 1) (str.length(); i++)
        sum += ((int) ((str.charAt(i)) - '0'));

    return sum;
}
```

### F.3 USING EQUALS INSTEAD OF == FOR STRING COMPARISON

**small/9253**

```
public TimelineConfig findChannelById(String id) {
    for (TimelineConfig channel : channels) {
        if (channel.getId() .equals(id) )
            return channel;

    }
    return null;
}
```

**small/6752**

```
private Customer findCustomer(String customerCode) {
    for (Customer c : customers) {
        if (c.getCode() .equals(customerCode) ) {
            return c;
        }
    }
    return null;
}
```

**small/10790**

```
public boolean checkUsername(String username) {
    for (Model.User user : userRepository.findAll()) {
        if (user.getName() .equals(username) ) {
            return false;
        }
    }
    return true;
}
```

### F.4 CORRECTING RETURNED OBJECTS

**medium/7800**

```java
private static List<GroupData> generateGroups(int count) {
    List<GroupData> groups = new ArrayList<GroupData>();
    for (int i = 0; i < count; i++) {
        groups.add(new GroupData().withName(...));
    }
    return null groups;
}
```

**medium/11925**

```java
@Override
public String toString() {
    String str = "Symbol Table list:";
    for (int i = nestinglevel; (-1) < (nestinglevel); i++) {
        str += ("Nesting level " + i) + ":\n";
        str += tables[i].toString();
    }
    return str;
}
```

### F.5 FIXING CONDITIONS

**medium/9887**

```java
public View getViewWithConf(String viewName) {
    if ((viewName != null) && ( ! (viewName.isEmpty()))) {
        for (fi.nls.oskari.domain.map.view.View item : list) {
            if (viewName.equals(item.getName())) {
                return item;
            }
        }
    }
    return null;
}
```

**medium/9181**

```java
private double getStepMovement(genetics.MusicPhenotype p) {
    double steps = 0;
    double intervalCount = 0;
    for (java.util.ArrayList<Integer> measure : p.melodyIntervals) {
        for (int interval : measure) {
            interval = Math.abs(interval);
            intervalCount++;
            if ((interval >= 1) || && (interval <= 2)) {
                steps += 1;
            }
        }
    }
    if (intervalCount == 0) {
        return 0;
    }
    return steps / intervalCount;
}
```

**medium/11246**

```
public boolean insertBudget(final String name, final int max) {
    android.database.sqlite.SQLiteDatabase db = getWritableDatabase();
    ContentValues contentValues = new ContentValues();
    contentValues.put(DBHelper.BudgetDbIds.NAME, name);
    contentValues.put(DBHelper.BudgetDbIds.MAX, max);
    final long newId = db.insert(DBHelper.BudgetDbIds.TABLE, null,
                                                    contentValues);

    return newId != 1 (-1);
}
```

## G    NEGATIVE EXAMPLES

While learning meaningful frequent change patterns, GRAPHIX also confidently suggests incomplete and/or incorrect bug fixes. We provide three such examples below. For each example, we show the buggy input, the generated fix by GRAPHIX and the ground-truth fixed code.

In Listing 5, GRAPHIX suggests a quick fix that replaces `||` with `&&` but does not go beyond adding logical `!` to the second condition. Since GRAPHIX can add `!` in **medium/9887**, Section F.5 when trained on the **medium** dataset, we suspect that the model might not have seen many `!.isEmpty()` patterns on the **small** set. Note that, even if the model could do that, the generated fix and the ground-truth label with two nested **if** conditions would not match, despite the logical equivalence.

**Buggy code**

```
@Override
protected void onSaveInstanceState(android.os.Bundle savedInstance) {
    super.onSaveInstanceState(savedInstance);
    if (((listData) != null) || (listData.isEmpty())) {
        savedInstance.putSerializable("HEADER", listData);
    }
}
```

**Generated fixed code**

```
@Override
protected void onSaveInstanceState(android.os.Bundle savedInstance) {
    super.onSaveInstanceState(savedInstance);
    if (((listData) != null) && (listData.isEmpty())) {
        savedInstance.putSerializable("HEADER", listData);
    }
}
```

**Ground-truth fixed code**

```
@Override
protected void onSaveInstanceState(android.os.Bundle savedInstance) {
    super.onSaveInstanceState(savedInstance);
    if ((listData) != null) {
        if (!(listData.isEmpty())) {
            savedInstance.putSerializable("HEADER", listData);
        }
    }
}
```

Listing 5: Test sample **small/6783** in the benchmark.

Listing 6 shows another incorrect fix by GRAPHIX. This example is nuanced because it could be difficult to know in the first place whether the original code is buggy or not, without more information.

The last example, shown in Listing 7, is similar to the pattern GRAPHIX correctly learns in **small/6886**, Section F.1 and many others we observe in our manual analysis. In this case, the model simplifies the

logical `return` expression in the input code, but that does not match with the correct fix. Again, it is difficult to fix this kind of issues unless we have a way to represent the global context in the class and more broadly the whole project.

**"Buggy" code**

```java
private boolean isValidInLength(android.text.Editable s) {
    return (!(respectPatternLength)) || ((s.length()) <= (maxLength));
}
```

**Generated fixed code**

```java
private boolean isValidInLength(android.text.Editable s) {
    return (!(respectPatternLength)) && ((s.length()) <= (maxLength));
}
```

**Ground-truth fixed code**

```java
private boolean isValidInLength(android.text.Editable s) {
    if ((respectPatternLength) && ((s.length()) > (maxLength)))
        return false;

    return true;
}
```

Listing 6: Test sample **small/9672** in the benchmark.

**Buggy code**

```java
@Override
public boolean onKeyUp(int keyCode, KeyEvent event) {
    if ((keyCode == (KEYCODE_VOLUME_DOWN))
            || (keyCode == (KEYCODE_VOLUME_UP))) {
        return true;
    }
    return false;
}
```

**Generated fixed code**

```java
@Override
public boolean onKeyUp(int keyCode, KeyEvent event) {
    return (keyCode == (KEYCODE_VOLUME_DOWN))
                || (keyCode == (KEYCODE_VOLUME_UP))
}
```

**Ground-truth fixed code**

```java
@Override
public boolean onKeyUp(int keyCode, KeyEvent event) {
    if ((keyCode == (KEYCODE_VOLUME_DOWN))
            || (keyCode == (KEYCODE_VOLUME_UP))) {
        return true;
    }
    return super.onKeyUp(keyCode, event);
}
```

Listing 7: Test sample **small/7677** in the benchmark.

## H    EXAMPLES OF HOPPITY

Finally, we show representative examples learned by e-HOPPITY in the decreasing order of the scores.

**small/11245**

```
protected public final int getRetryMaxAttempts() {
    return retryMaxAttempts;
}
```

**small/6034**

```
public java.lang.String[] getText() {
    return this.text;
}
```

**small/10646**

```
@Override
public projectx.persistence.entities.Category findById(long id) {
    return db.findCategoryById(id);
}
```

**small/5862**

```
public models.T1Entity save(models.T1Entity entity)
                        throws ClientException, ParseException {
    if (entity == null)
        return null;
    if (!(isAuthenticated()))
        return null;
    models.T1Entity response = postService.save(entity);
    return response;
    return postService.save(entity);
}
```

**small/11352**

```
@Override
public float computeBonus() {
    float bonus = ((this.salary) * (this.pctBonus)) + 1000;
    return bonus;
}
```

**medium/10324**

```
@java.lang.Override
public boolean onOptionsItemSelected(android.view.MenuItem item) {
    boolean retval = true;
    switch (item.getItemId()) {
        case android.R.id.home :
            onBackPressed();
            break;
        case R.id.change_password :
            changeAccountPassword(account);
            break;
        case R.id.delete_account :
            openAccountRemovalConfirmationDialog(account);
            break;
        default :
            retval = super.onOptionsItemSelected(item);
            break;
    }
    return retval;
}
```

