# OpenReview forum: "GRAPHIX: A Pre-trained Graph Edit Model for Automated Program Repair"
_ICLR.cc/2022/Conference — ICLR 2022 Submitted_

### Official Review · Reviewer_AZoW · 2021-10-24

**Correctness:** 3
**Technical Novelty And Significance:** 2
**Empirical Novelty And Significance:** 2
**Recommendation:** 5
**Confidence:** 5

**Main Review:**

Strengths:
+ The problem formulation of mapping a graph to a series of AST edits lends itself naturally to modeling source code, and repairs in particular.
+ Incorporating a large set of baselines and variations helps contextualize Graphix' performance.

Weaknesses:
- Contributions over prior work are relatively slim and hard to assess because the work lacks a comprehensive comparison.
- The efficacy of the proposed multihead GNN scheme is not adequately ablated.
- Some concerns about the scalability given the need to encode the entire tree at each intermediate step.

This work explores a promising direction, injecting richer inductive bias into neural models of source code and its edits. The scope of its technical contributions is rather narrow and although the paper includes a fair set of comparisons and ablations, their practical usefulness is difficult to assess because several key comparisons are missing. Please focus on these in the response/future revisions.

The first main contribution over Hoppity is the use of a multi-head scheme in the GNN encoder. While the name might suggest otherwise, this is an ensemble of 8 independently trained GNNs whose final states are concatenated. This naturally increases the parameter cost of the GNN by a factor 8, which requires ablations with models of the same size. The paper only offers a few such comparisons, all of which compensate by equipping the non-multi-head model with a larger hidden dimension, which is somewhat naturally prone to overfitting. This really calls for another variant that instead increases the layer count; that number is set to a surprisingly low number (4) in this work, and it seems entirely plausible that using an ensemble would compensate for this depth rather than width. This hypothesis needs to be tested. Also consider combinations of increasing both depth and width.

The second contribution is the ASDL-guided decoder. This aspect is only ablated through comparison with a small instantiation of Hoppity in Table 5 (E.2., where it should be made more clear that GRAPHIX-B refers to a single-headed model). This points to a larger problem: Hoppity is never analyzed with more than 1M parameters, despite Graphix using 32M by default. Comparisons with Graphix-B (also using 1M parameters) need not translate to the higher parameter domain. This is especially relevant since the multi-head idea discussed earlier does not seem quite as impactful, perhaps especially when properly ablated, nor does the pretraining (as evidenced later), as the ASDL contribution here. A similar problem is present in E.3. where the per-layer and last-layer performance is virtually identical in the middle two rows, so why was per-layer not evaluated at 32M parameters? (And why only using averaging?). On a smaller note, the runtime cost of operating Graphix vs. Hoppity should be reported and considered as well.

Overall, the paper uses a natural approach to this task, but much of it is similar to prior work, and the new components feel inadequately analyzed. Please provide at least equal comparisons of Hoppity and one or more deeper non-multihead Graphix models in the 32M parameter domain. Besides these salient issues, several more problems, some minor and others moderately significant, are identified below.

Detailed Comments:
- The first contribution on page 2 seems to claim a very specific "first". Was the goal to claim it as the "first medium-scale graph edit model" and list the rest of the sentences as its properties?
- The motivational example is rather generic and does not really articulate why Graphix should work better than other edit-based methods. That edits (and graphs) are a better formalism has been shown quite extensively in prior work (e.g., Ding et al., ASE'20, or many of the cited works).
- The idea of using multiple heads in graph encoders is naturally reminiscent of Transformers, and more specifically of relationally biased attention such as proposed in RAT (Wang et al., ACL'20) or GREAT (Hellendoorn et al., ICLR'20). This probably deserves a comparison.
- Reinstantiating and encoding the program graph on every edit would seem to present major scalability problems, especially on longer chains of edits. Please analyze and discuss.
- It is not quite clear whether the ensemble-of-graphs model is meant to be a contribution in general or just for source code; the former would naturally require more analysis on different datasets, but the latter probably calls for a discussion of why we expect this to just/primarily be relevant for code.
- The contribution of pretraining seems quite slim; performance only increases a little on the Small dataset and actually decreases on the Medium data. The latter probably deserves more analysis if this is to be considered a part of the contribution of this work: did you attempt to pretrain with removing larger subtrees?
- The introduction and conclusion decidedly overpromise on performance, claiming that Graphix performs well and is 10x smaller than recent models -- those recent models also significantly outperform it in many settings, and models like PL-BART are just 4 times larger than Graphix (much less than the 32x difference of Graphix with Hoppity).
- The examples in section 6 suffer from the same problem as the motivation: they do not shed any light on why the specific additions of Graphix over comparable models are helpful.
- Why are most baselines in Table 1 not evaluated on the concrete setting?

**Summary Of The Paper:**

Automated program repair benefits from knowledge of its many properties, which includes its inherent (parse-)tree structure and graphical properties such as data-flow. This work proposes a graph-based encoder coupled with a tree-edit decoder, and optionally pretrained on a tree-based objective comparable to masked language modeling. The resulting model efficiently leverages relatively few parameters to achieve near-SOTA performance on a benchmark compiled from real-world bug fixes.

**Summary Of The Review:**

The work proposes two modeling contributions over prior work, neither of which is ablated quite carefully enough, primarily in terms of comparing models of more reasonable size. The eventual results are not particularly strong compared to many other baselines, which, while equipped with more parameters, also use models that scale better with larger parameter budgets.

---

> ### Author Response · Authors · 2021-11-23
> **Response to Reviewer AZoW (1)**
>
> Thank you for the detailed review and thoughtful comments.
>
> We first address your main concerns regarding the performance of Graphix-P on the medium data, the scalability aspect and the efficacy of multi-head GNNs. We then answer the remaining questions.
>
> ### 1. Graphix-P performance on the medium data
>
> > "The latter probably deserves more analysis if this is to be considered a part of the contribution of this work: did you attempt to pretrain with removing larger subtrees?"
>
> Thanks for the suggestion. As you and *Reviewer V4AB* suggested, we try with larger sub-trees from 3 to 8 nodes and continue training the existing pre-trained model. Note that we choose 8 nodes at maximum due to a scalability issue that we address in the response to Reviewer AZoW. We simplify the sequence of ADD operations with a mixed of ADD and COPY in the synthetic edits if there is a matched sub-tree found in other parts in the AST.
>
> We pre-train the model for 2 more epochs and fine-tune Graphix again. The results are shown in the following Table where we see some improvements on both small and medium datasets. The results suggest that pre-training with larger sub-trees (typically 15% corruption rate in pre-training Transformer models) may help further.
>
> | Pre-trained | Model | Model Size | Small | Medium |
> | ----- | ----- | :-----: | ----- | ----- |
> | No | GRAPHIX | 32M | 18.20% | 9.19% |
> | Sub-tree of 2-6 nodes | GRAPHIX-P | 32M | 19.81% | 8.81% |
> | Sub-tree removal of 2-8 nodes + COPY op | GRAPHIX-P | 32M | 19.92% | 9.56% |
>
>
> ### 2. Scalability issue
>
> > *"Reinstantiating and encoding the program graph on every edit would seem to present major scalability problems, especially on longer chains of edits. Please analyze and discuss."*
>
> Thank you for raising this concern. We agree with your assessment and appreciate your willing to discuss further.
>
> As you mentioned, at each step we previously reinstantiated the graph (i.e., constructing the program graph from the AST, adding additional value nodes and links) and ran the graph encoder again. This led to both running time and memory overhead compared to sequence models, and limited us from training larger models and trying with larger sub-trees for pre-training.
>
> To address this, we experiment with a more light-weight encoding scheme, which is similar to the sequence processing with RNN. The overall idea is to only re-compute the node embeddings when there is a significant change to the graph. More specifically,
>
> 1. For node addition, we only compute the embedding of the new node by taking its node type embedding and the embedding of its parent node from the previous step. We do that by constructing the program graph with a reduced node and edge lists as input to the GNN. We do the same with the copy operation, which adds a sub-tree.
> 2. For node removal and value replacement, we reinstantiate the graph encoding again, just like we originally do.
>
> This allows us to reduce the memory by 2x and the running time slightly. Effectively with a larger batch size, which was 10 per GPU for small and 5 per GPU for medium before, we can speed up the training. We can also pre-train the model by removing larger sub-trees.
>
> We see slightly better results with the new encoding as shown in the below Table
>
> | Encoding | Model	| Pre-trained	| Size	| Small	| Medium	| Per epoch runtime small/medium |
> | ----- | ----- | :-----: | ----- | ----- | ----- | :-----: |
> | Old	| GRAPHIX	| No	| 32M	| 18.20%	| 9.19% | 	1h/3.5h |
> | Simplified encoding	| GRAPHIX	| No	| 32M	| 18.85% | 	9.29%	| 0.6h/1.65h |
>
> The scalability can further be improved by:
>
> 1. Using edge embeddings instead of multiple graphs, one per edge type;
> 2. Allocating a fixed size for the tree nodes (says, 500) and the value node to avoid re-indexing the graph nodes after each decoding step; and
> 3. Keeping the same latent dimension in the decoder as the encoder.

---

> > ### Comment · Reviewer_AZoW · 2021-11-25
> > **Discussion of Author Response**
> >
> > Thank you for your very detailed response to my comments. I discuss these point-by-point below.
> >
> > 1. This is indeed somewhat encouraging. It would be good to see an ablation that only expands the subtree size, without adding the COPY op change as well, but I recognize that time constraints played a role in this decision.
> > 2. The simplified encoding scheme looks like a significant and interesting addition.
> > 3. Thanks for including this comparison. It does indeed look like the ensemble/multi-head model makes no substantial contribution over a simpler architecture. I would recommend against including it as a contribution, at least barring either further analysis on other domains or evidence of a substantial & significant improvement in performance. Mentioning it as an ablation or alternative instantiation seems fine.
> > 4. I appreciate the discussion of Hoppity vs. Graphix. The substantially shorter edit sequence length definitely seems like a worthwhile point. Although there is no significant decrease in runtime, it seems quite reasonable to argue that this is the primary factor for Graphix' scaling much better to medium-length sequences.
> >
> > The bigger picture is that the response discusses some pretty substantial revisions to the paper -- introducing a new encoding scheme, adding ablations and different style pretraining, discussing limitations of the multi-head graph encoder. Revising the paper (significantly) with this in mind would certainly strengthen it. Yet, nearly none of these changes are reflected in the paper; only some smaller edits were made by the revision deadline. For this reason, I raise my score, but do not recommend acceptance of the manuscript in its current form. I do strongly encourage the authors to pursue the promising identified revisions for a future revision.

---

> > > ### Author Response · Authors · 2021-11-26
> > > **Response to Reviewer AZoW**
> > >
> > > Thank you for the constructive feedback and suggestions. As authors, we find your insightful review and response is very helpful in improving our work. Based on the reviews and your recent response, we have made the following changes to your paper:
> > >
> > > * Describe the simplified encoding scheme and the results on scalability;
> > >
> > > * Include the ablation results;
> > >
> > > * Deemphasize of the multi-head architecture as a contribution.
> > >
> > > The specific details of these changes are the following:
> > >
> > > ### Approach
> > >
> > > We moved the graph encoder description in Appendix A.1 to the beginning of Section 4.1 and discussed two main architectural differences of the graph encoder with Hoppity:
> > >
> > > 1) **Multi-head model:** we revised the paragraphs about the multi-head encoder (previously section 4.1) and deemphasized its importance as a new idea/contribution. We described the multi-head architecture as an alternative to depth and width of the graph network in increasing the model capacity, which is crucial in establishing the results as competitive as the Transformer baselines.
> > >
> > > 2) **Simplified graph encoding:** we discussed the scalability issue of reinstantiating the graph at each step of the decoding process. Then, we described a simplified encoding scheme to partly overcome this issue.
> > >
> > > ### Experiments
> > >
> > > We only kept some details of Sections 5.1 and 5.2 and moved the remaining to Appendix, per the suggestion of Reviewer z3GX.
> > >
> > > We dedicated a new section to make a clear comparison between Graphix and Hoppity. We included the results, as shown in our response, that compare the two approaches in terms of the impact of shorter edit sequences, the ASDL-decoding strategy and the model sizes.
> > >
> > > We also discussed the runtime cost of both approaches and provided more ablation results on incremental decoding and of the multi-head component, and of increasing depth and width.
> > >
> > > The original section 5.4 is now devoted to comparing Graphix with the sequence-based (pre-trained) baselines. Here, we discussed Graphix’s (non-) pre-training as well as ablation studies in terms of the size of corrupted sub-trees.
> > >
> > > Note that these new sections focus on the abstract dataset, so we use the next section to discuss the remaining results on the concrete code and why such results are important.
> > >
> > > In summary, these changes only involve clarifications, reorganizations and additions of text and ablation results, hence we do not substantially change the submission.
> > >
> > > Please let us know if you have further suggestions on the revision.

---

> ### Author Response · Authors · 2021-11-23
> **Response to Reviewer AZoW (2)**
>
> ### 3. Efficacy of multi-head GNNs
>
> > *"This really calls for another variant that instead increases the layer count; that number is set to a surprisingly low number (4) in this work, and it seems entirely plausible that using an ensemble would compensate for this depth rather than width. This hypothesis needs to be tested. Also consider combinations of increasing both depth and width."*
>
> Thanks for the insight. Our ablation study in Appendix E.2. suggests that merely increasing the width leads to overfitting. Increasing the GNN depth is a direction we did not explore originally. For that, we run another experiment with $H=1, L=8$ and $D=512$, which has about 32M parameters.
>
> We report the results in the following table and compare with the multi-head model. Both are using the new simplified graph encoding. It turns out that increasing both depth and width does help the model perform as well as the multi-head version.
>
> | Model    | Heads    | Width    | Depth    | Model size    | Small    | Medium    | Per epoch runtime (small/medium) |
> | ----- | ----- | ----- | ----- | ----- | ----- | ----- | ----- |
> | GRAPHIX    | 8    | 256    | 4    | 32M    | 18.85%    | 9.29%    | 0.6h/1.65h |
> | GRAPHIX    | 1    | 512    | 8    | 32M    | 18.27%    | 8.93%    | 0.4h/1h |
>
>
> ## 4. Remaining questions
>
> > *"It is not quite clear whether the ensemble-of-graphs model is meant to be a contribution in general or just for source code; the former would naturally require more analysis on different datasets, but the latter probably calls for a discussion of why we expect this to just/primarily be relevant for code."*
>
> That is a great question. From the experiments on the depth above, it seems that an ensemble of graphs is an alternative to depth and width in increasing the model capacity. We do not know how impactful the ensemble idea is in general.
>
> > *"The second contribution is the ASDL-guided decoder. This aspect is only ablated through comparison with a small instantiation of Hoppity in Table 5 (E.2., where it should be made more clear that GRAPHIX-B refers to a single-headed model). This points to a larger problem: Hoppity is never analyzed with more than 1M parameters, despite Graphix using 32M by default. Comparisons with Graphix-B (also using 1M parameters) need not translate to the higher parameter domain. This is especially relevant since the multi-head idea discussed earlier does not seem quite as impactful, perhaps especially when properly ablated, nor does the pretraining (as evidenced later), as the ASDL contribution here."*
>
> Thanks for the question. Before the submission, we did train such a Hoppity model with 32M parameters on the medium data. The result got improved from 1.21% to 1.57% but still too low for us pursue it further.
>
> Three factors contribute to the improvement over Hoppity: (1) the use of shorter and more meaningful edits, demonstrated in Appendix B; (2) the grammar-guided decoding; (3) the multiple-head encoder as a way to increase the model capacity and a new pre-training strategy.
>  The following table below illustrates these aspects:
>
> | Model | Edit | Avg. small length | Avg. medium length| Model size | Small | Medium |
> | ------ | ----- | :-----: | :-----: | ----- | ----- | ----- |
> | HOPPITY | JSON diff edit | 14.1 | 35.2 | 1M | 7.30% | 1.21% |
> | HOPPITY | JSON diff edit | 14.1 | 32.2 | 32M | N/A | 1.57% |
> | HOPPITY | JSON diff edit + COPY | 7.3 | 12.6 | 1M | 8.47% | 2.91% |
> | GRAPHIX |Grammar-aware edit | 5.4 | 8.5 | 1M | 12.40% | 5.34% |
> | GRAPHIX |Grammar-aware edit | 5.4 | 8.5 | 32M | 19.92% | 9.56% |
>
> In short, our work is a result of empirical insights that help achieves near state-of-the-art results using a paradigm that is amenable to more program structures.
>
> > *"A similar problem is present in E.3. where the per-layer and last-layer performance is virtually identical in the middle two rows, so why was per-layer not evaluated at 32M parameters? (And why only using averaging?). On a smaller note, the runtime cost of operating Graphix vs. Hoppity should be reported and considered as well."*
>
> Accumulating the per-layer head outputs increases the latent dimension at each later layer and leads to more computational burden, so we used the averaging for simplicity. However, we observed that that averaging the head outputs (in 32M model) works as well as concatenation. We didn’t not evaluate the per-layer architecture at 32M parameters for the same computational reason.
>
> In terms of running time, with the same number of edits steps, Graphix runs as fast as Hoppity since the ASDL-based decoding introduces negligible complexity. Please note that the average edit length of JSON differencing used by Hoppity are 14.1 and 35.2 for the small and medium datasets, compared to 5.4 and 8.5 respectively for Graphix. To learn complex edits, Hoppity would have to handle much longer JSON diff edits and hence incur more runtime cost.

---

> ### Author Response · Authors · 2021-11-23
> **Response to Reviewer AZoW (3)**
>
>
> > *"The idea of using multiple heads in graph encoders is naturally reminiscent of Transformers, and more specifically of relationally biased attention such as proposed in RAT (Wang et al., ACL'20) or GREAT (Hellendoorn et al., ICLR'20). This probably deserves a comparison."*
>
> Thanks for the references. We will add and discuss them in our revision.
>
>
> > *"The first contribution on page 2 seems to claim a very specific "first". Was the goal to claim it as the "first medium-scale graph edit model" and list the rest of the sentences as its properties?"*
>
> We have changed the wording of this sentence.
>
>
> > *"The introduction and conclusion decidedly overpromise on performance, claiming that Graphix performs well and is 10x smaller than recent models -- those recent models also significantly outperform it in many settings, and models like PL-BART are just 4 times larger than Graphix (much less than the 32x difference of Graphix with Hoppity)."*
>
> Thank you for pointing out the inaccuracy of our claim about the model size. This was a mistake! We have corrected the claim in the revision with some improved results. In short, our model is about 2x smaller than CodeT5-small and perform better on the small data.
>
> > *"The examples in section 6 suffer from the same problem as the motivation: they do not shed any light on why the specific additions of Graphix over comparable models are helpful."*
>
> Besides the performance in accuracy, the anecdotal examples qualitatively show the kinds of bugs Graphix is able fo fix for the particular test benchmark since our goal is to understand whether or not Graphix can be used in real-world applications. For practical use, the precision should be high and the fixes should be “interesting”. Table 4, Appendix D addresses the former while the anecdotal examples gives the grounds for the latter. We will clarify that the anecdotal examples are not intended for comparison.
>
>
> > *"Why are most baselines in Table 1 not evaluated on the concrete setting?"*
>
> It is because the published benchmark uses abstract code, which was introduced by Tufano et al., 2019 to deal with the open vocabulary issues. Only DeepDebug (Drain et al., 2021) started to work on concrete code recently.

---

### Official Review · Reviewer_V4AB · 2021-11-01

**Correctness:** 3
**Technical Novelty And Significance:** 2
**Empirical Novelty And Significance:** 2
**Recommendation:** 6
**Confidence:** 4

**Main Review:**

### Pros

- The paper tackles the task of program repair, which is relatively new and can have practical impact. Although solving a very hard problem, the work is able to outperform various baselines and have on-par performance with very large models.
- The proposed architecture is able to generate longer graph edits which can inherently fix more programs.
- Empirical study covers many recent works and is fairly comprehensive.
- The anecdotal examples as well as the examples provided in the Appendix are indeed interesting.

### Cons

Major

A. Although claimed to be a novel pre-training task, it is not THAT novel. Moreover, although "masking tokens" is natural in pre-training sequence encoders, "masking sub-trees" on AST is essentially just asking the model to do "missing code prediction", encompassing only a small subset of general program repair samples. The pre-training task would be far more "novel" if more thoughts has been put into generating the pre-training dataset. For example, how to generate data for pre-training other edits such as copy, remove, and update.

B. Empirically, pre-training is costly (according to the Appendix) but not giving a lot of performance gain (<2%). The problem is much worse on medium sized data. Here are two questions:
	 1. Why is pre-training only done on "sub-trees between 2 and 6 descendants"? How about increasing the size in order to cover longer edit sequences during pre-training? Is it possible to add such an experiment?
	 2. (Related to A) Is it because the pre-training task only asks the model to predict a sequence of addition operations? Can we add more diverse tasks to let the model to predict a variety of edits?

C. Multi-head graph encoder does not sound too novel to me.


Other Issues

. Miss related works:

TFix: Learning to Fix Coding Errors with a Text-to-Text Transformer

Learning semantic program embeddings with graph interval neural network

. The last paragraph in Section 4.1 shows that there is a single-head "base" model, which is not mentioned in any experiment in the main paper. I found the "base" in Appendix but it would be better if you move this description into Appendix as it is not that relevant. Alternatively you can also put the "base"  model and its performance inside Table 1/2.

. Listing 2 Top: the code is not valid if the highlighted green part is removed. What is actually the before and after of that code snippet?

. The paper claims "these observations suggest that the code abstraction may not be necessary". Despite finding that the argument is pretty weak, I also don't think "code abstraction is necessary" in the first place. For example, with abstraction one would not find the fix for Listing 3 (bottom example).

### Nitpicks

- Highlight colors (green and red) used in code snippets are too intense. Can be toned down for ease of read
- Listing 2 Bottom: trailing whitespace is also included in the highlight.
- Table 1 and 2 are hard to decode. Maybe try separating them into multiple tables (with pre-train and without pre-train).

**Summary Of The Paper:**

The paper presents GRAPHIX, a graph edit model for program repair. The work is directly related to Hoppity (Dinella et. al. 2020) which proposed using a sequence of graph edit for program repair. GRAPHIX employs multi-head graph encoder which improves upon Hoppity in terms of accuracy and complexity. Notably GRAPHIX is able to learn longer edit sequence and thus work on more program repair samples. The work has also proposed a pre-training task to improve model performance. Empirically the authors evaluated GRAPHIX on the *Patches in the Wild* Java bug-fix benchmark. It outperforms various baselines without pre-training. With pre-training, GRAPHIX-P stays roughly on par despite having much smaller model.

**Summary Of The Review:**

Overall, I think the paper has merits. Program repair in the wild using end-to-end neural network is inherently a difficult problem. The proposed architecture shows a good performance while using a much smaller model, and can spawn further discussion of algorithms using graph edits for program repair. The paper has also shown a pretty comprehensive evaluation. However, the paper will be in a much better shape if the pre-training task is designed more throughly. We would also hope there is a better performance gain from incorporating pre-training. I lean towards accepting the paper.

---

> ### Author Response · Authors · 2021-11-23
> **Response to Reviewer V4AB**
>
> Thank you for your detailed comments and suggestions. We answer your questions below.
>
> > *"Empirically, pre-training is costly (according to the Appendix) but not giving a lot of performance gain (<2%). The problem is much worse on medium sized data. Here are two questions: 1. Why is pre-training only done on "sub-trees between 2 and 6 descendants"? How about increasing the size in order to cover longer edit sequences during pre-training? Is it possible to add such an experiment? 2. (Related to A) Is it because the pre-training task only asks the model to predict a sequence of addition operations? Can we add more diverse tasks to let the model to predict a variety of edits?"*
>
> We selected sub-trees between 2 and 6 nodes for pre-training under the assumption that the synthetic data should mimic the edit length distribution of the labelled datasets (Table 1, Appendix B). However, the pre-training did not give a performance gain on the medium data.
>
> As you and *Reviewer AZoW* suggested, we try with larger sub-trees from 3 to 8 nodes and continue training the existing pre-trained model. Note that we choose  8 nodes at maximum due to a scalability issue that we address in the response to Reviewer AZoW. We simplify the sequence of ADD operations with a mixed of ADD and COPY in the synthetic edits if there is a matched sub-tree found in other parts in the AST.
>
> We pre-train the model for 2 more epochs and fine-tune Graphix again. The results are shown in the following Table where we see some improvements on both small and medium datasets. The results suggest that pre-training with larger sub-trees (typically 15% corruption rate in pre-training Transformer models) may help further.
>
> | Pre-trained | Model    | Model Size    | Small    | Medium |
> | ----- | ----- | :-----: | ----- | ----- |
> | No | GRAPHIX | 32M    | 18.20%    | 9.19% |
> | Sub-tree of 2-6 nodes    | GRAPHIX-P    | 32M    | 19.81%    | 8.81% |
> | Sub-tree removal of 2-8 nodes + COPY op    | GRAPHIX-P    | 32M    | 19.92%    | 9.56% |
>
>
> > "Multi-head graph encoder does not sound too novel to me."
>
> We agree that the proposed multi-head encoder is one way to provide the model with an additional capacity. But it is an important component of the overall graph edit model that achieve near state-of-the-art results compared to large-scale Transformer models.
>
> > "Miss related works"
>
> Thank you for the reference. We have added them in our revision.
>
> Finally, the new results are on the abstract dataset. We will obtain the results on the concrete code and include all in our revision. We will fix other formatting issues as you suggest.

---

> > ### Comment · Reviewer_V4AB · 2021-11-29
> > **Acknowledgement of author response**
> >
> > I thank the authors for the detailed response. Overall, given the changes the author have made, I think the paper can be accepted though I agree the work is bit weak on the novelty front.

---

### Official Review · Reviewer_iXHo · 2021-11-02

**Correctness:** 3
**Technical Novelty And Significance:** 2
**Empirical Novelty And Significance:** 2
**Recommendation:** 5
**Confidence:** 5

**Main Review:**

## Comments
* It is unclear to me what it means that the state of the art was not beaten. I am missing a discussion on why the presented work is still relevant despite that.
* The anecdotal examples are not compared to the kind of results other approaches give, so it is unclear what to make of them.
* Repeatedly, the claim is made that the model is more than 10x smaller than “current large scale sequence models” but out of the evaluated baselines only “BART” is that much larger. The presented approach is even outperformed by CodeT5-small, which is less than 2x as large.
* I would have liked a short discussion on how accurate you think it is to detect bug fixes by filtering for “fix”, “bug”, “error” and ”exception” (page 6).
* “To build the final dataset, the authors extracted about 10M GitHub commits whose messages textually match one of these (“fix”, “bug”, “error”, and “exception”) patterns.” -- Remove “,” before the “and” (page 6)
* What is the reasoning behind having exactly 600 nodes (page 7)?
* Table 1 might benefit from a visual separation between edit based and sequence based approaches, maybe using color?
* “Additionally, we notice that while the pre-training provides an additional 10% relative gain on the small subsets, GRAPHIX-P does not achieve similar gains on the medium set, which is inherently longer and harder.” Seems a bit euphemistic as the results got slightly worse.


**Summary Of The Paper:**

The paper proposes a new approach to abstract syntax tree-based automatic program repair. The novel technique called deleted-subtree reconstruction is based on dropping parts of the syntax tree and training the model to grow them back. The method is evaluated against edit-based and sequence-based approaches, where it outperforms only the edit-based ones.

**Summary Of The Review:**

## Strengths:
* very well written and clearly structured text
* good motivating example
* interesting approach in scope of the conference
* gives a concise intro to previous work and ties back throughout the paper
* achieves respectable performance at comparable small model size

## Weaknesses:
* underperforms in terms of the state of the art
* anecdotal examples offer very limited insights in current form
* approach mostly orchestrates existing architectural concepts with little actual novelty

---

> ### Author Response · Authors · 2021-11-23
> **Response to Reviewer iXHo**
>
> Thank you for the review. We address your concerns below.
>
> > *“It is unclear to me what it means that the state of the art was not beaten. I am missing a discussion on why the presented work is still relevant despite that.”*
>
> Though our work does not beat the state-of-the-art results, we present a competitive approach for program repair from a different program representation. We have discussed in the introduction the motivation behind the edit-based approach, and the importance and potential of incorporating program structures.
>
>
> > "The anecdotal examples are not compared to the kind of results other approaches give, so it is unclear what to make of them."
>
> We will clarify that the anecdotal examples are not intended for comparison. However, besides the performance in accuracy, the anecdotal examples qualitatively show the kinds of bugs Graphix is able fo fix for the particular test benchmark since our goal is to understand whether or not Graphix can be used in real-world applications. For practical use, the precision should be high and the fixes should be “interesting”. Table 4, Appendix D addresses the former while the anecdotal examples gives the grounds for the latter.
>
> > *“Repeatedly, the claim is made that the model is more than 10x smaller than “current large scale sequence models” but out of the evaluated baselines only “BART” is that much larger. The presented approach is even outperformed by CodeT5-small, which is less than 2x as large."*
>
> Thank you for pointing out the inaccuracy of our claim about the model size. This was a mistake! We have corrected the claim in the revision. In short, our model is about 2x smaller than CodeT5-small, performing better on the small data.
>
> > *“I would have liked a short discussion on how accurate you think it is to detect bug fixes by filtering for “fix”, “bug”, “error” and ”exception” (page 6).“*
>
> As we briefly discuss in the Introduction, paragraph 2, curating a reasonably large data set for program repair (and beyond the method changes) is challenging, so extracting bug-fix data by filtering code changes with “fix”, “bug”, “error” and ”exception” keywords is a heuristic, which is not as accurate. In fact, the benchmark itself is noisy as well.
>
> > *"What is the reasoning behind having exactly 600 nodes (page 7)?"*
>
> The length of abstract code in the medium data is between 50 and 100 after the abstraction and idiomization step (e.g., converting  `System.out.println` to a simple name `IDIOM_1`). Without that step, the concrete code is about 3x longer on average, so parsing the concrete code into AST requires at least 500 nodes to cover every sample. Since we also keep dummy nodes for unfilled optional children (e.g., having a dummy node for `elseStmt` when there is not an else statement), we choose 600 nodes. For reference, HOPPITY uses 500 nodes on JavaScript.
>
> > *"Table 1 might benefit from a visual separation between edit based and sequence based approaches, maybe using color?"*
>
> Thank you. We will do that in the revision.
>
> > *“Additionally, we notice that while the pre-training provides an additional 10% relative gain on the small subsets, GRAPHIX-P does not achieve similar gains on the medium set, which is inherently longer and harder.” Seems a bit euphemistic as the results got slightly worse."*
>
> Thanks for raising this question. As Reviewer V4AB and Reviewer AZoW suggested, we try with larger sub-trees from 3 to 8 nodes and continue training the existing pre-trained model. Note that we choose  8 nodes at maximum due to a scalability issue that we address in the response to Reviewer AZoW. We simplify the sequence of ADD operations with a mixed of ADD and COPY in the synthetic edits if there is a matched sub-tree found in other parts in the AST.
>
> We pre-train the model for 2 more epochs and fine-tune Graphix again. The results are shown in the following Table where we see some improvements on both small and medium datasets. The results suggest that pre-training with larger sub-trees (typically 15% corruption rate in pre-training Transformer models) would help further.
>
> | Pre-trained | Model    | Model Size    | Small    | Medium |
> | ----- | ----- | :-----: | ----- | ----- |
> | No | GRAPHIX | 32M    | 18.20%    | 9.19% |
> | Sub-tree of 2-6 nodes    | GRAPHIX-P    | 32M    | 19.81%    | 8.81% |
> | Sub-tree removal of 2-8 nodes + COPY op    | GRAPHIX-P    | 32M    | 19.92%    | 9.56% |
>
> Finally, the new results are on the abstract dataset. We will obtain the results on the concrete code and include all in our revision.

---

### Official Review · Reviewer_z3GX · 2021-11-03

**Correctness:** 3
**Technical Novelty And Significance:** 2
**Empirical Novelty And Significance:** 3
**Recommendation:** 5
**Confidence:** 3

**Main Review:**

This is an ambitious paper that tries to pack a lot in: between the main paper and the appendix, the authors introduce a (non-trivial, graph-aware) model; a pre-training regime for the code repair problem; quantitative evaluations against many benchmark models; and qualitative examples of some code repair predictions. I really do admire and appreciate the effort that went into this work, which seems significant!

However, the net effect is of many, many parts to this paper without a convincing demonstration of model novelty or of significant improvement to the SOTA on the provided baseline. I provide more details below:

Strengths:
1. Important and interesting application. Program repair is clearly useful, and the structured task also bears close relation to other tree and graph structured tasks (knowledge graph and structured schema editing; protein editing; language and grammar correction) that are clearly of interest to many.
2. Model is structure aware and predicts graph edits - an interesting choice with downstream possibilities. The authors present comparisons both to SOTA models that are not generally graph-aware (code pre-trained versions of BERT, Transformers, and T5) and at least two explicitly graph/AST-based models (GraphCodeBERT , HOPPITY). Unlike all the models evaluated except HOPPITY, the authors model is both structure-aware and predicts explicit edits to graphs, which feels like an interesting feature that may allow it to integrate more easily with other, non-neural systems (like a linter). It also outperforms the HOPPITY baseline by a large margin.
3. Extensive evaluation efforts against many comparable models. The authors compare their model against many SOTA models on two versions of the same dataset; and provide additional ablations on several parts of the model (eg. number of graph encoder heads) in the appendix.

Weaknesses and/or concerns:
1. Model novelty. The model architecture seems very closely based on the HOPPITY baseline: the primary change seems to be to the multi-head graph encoder (which itself uses an existing GIN network for each head). While it’s exciting that this addition improves over the original, the resulting model therefore doesn’t seem to introduce a fundamentally new idea: it also encodes graphs, and predicts edits to them. As a presentation nit, the bulk of the model architecture itself (beyond the ‘multi-head’ change) is described in the appendix.
2. Quantitative evaluations show marginal improvement at best in comparison to the large pre-trained models. I’d further break down some comments on the qualitative results section:
* The authors heavily emphasize that the ‘smaller number of parameters’ of their model in comparison to the best-performing SOTA. This feels marginally interesting -- and the ‘CodeT5-small’ model seems to perform very similarly to the model in this paper, with about the same order of magnitude of parameters. In fact, all of the best performing pre-trained models seem to perform about the same.
* The evaluation omits a comparison to the closely related Yao 2021 model that is cited throughout the paper.
* The pre-training regime, which is presented as a contribution of this paper, does not seem to produce markedly stronger results. Further, for a fair comparison, it might seem valid to apply the same pre-training regime to some of the other models (like HOPPITY; or the Yao 2021 model if that was added.)
* Presentationally, a good deal of space is taken up by discussing a quirk of the particular dataset evaluation chosen (the abstract vs. concrete versions of the dataset.) As it turns out, the performance of all of the better-performing pre-trained models seems to be about the same between these. This is useful knowledge for future users of this dataset, but not particularly enlightening for this paper. It seems like this space might have been better used with evaluation on another code repair dataset or domain.

3. Qualitative evaluation is intriguing but doesn’t yield clear conclusions. In particular, I’d be looking for evidence that ‘GRAPHIX demonstrates strong inductive biases in learning complex bug-fix patterns’ - or otherwise, some conclusion on the kinds of bugs that GRAPHIX tends to fix (largely syntactic or catchable by a linter? Requiring multiple edits in graph-space? Concentrated on specific kinds of operators?). One possibility is hand-coding the analyzed edits to categorize them and summarizing the results: without that, these examples are interesting but inconclusive.

4. Pre-training regime is not particularly novel. The authors suggest deleting elements of subtrees from existing code and reconstructing them. This is perhaps given more space in the paper than necessary, and offered as a contribution. The significance of this is a matter of reviewer opinion; I did not personally find it sufficiently novel.


**Summary Of The Paper:**

This paper presents a model over sequential structural tree edits, used for program repair based on ASTs. The model itself uses a graph encode and decoder to predict the sequential tree edits. The authors also introduce a method for pretraining the model on existing (non program repair) code data: they delete subtrees of arbitrary size from the code, and predict their reconstruction. The resulting model performs comparably to several other state of the art code repair models on the Patches in the Wild Java repair dataset, but with fewer parameters than several of the best-performing pre-trained models (CodeBERT, CodeT5).


**Summary Of The Review:**

In short, while I appreciate the significant effort that went into this paper, I found it comprised of many parts that fell just short of convincing execution. The model architecture presents only minor changes over an existing one; the quantitative evaluations show only minor improvements over the SOTA (and the authors rely heavily on the qualified claim that they are the best performing ‘mid sized model’ to distinguish themselves, a claim only somewhat supported, given the t5-small model); and present inconclusive and hard to interpret qualitative results.

---

> ### Author Response · Authors · 2021-11-23
> **Response to Reviewer z3GX**
>
> Thank you for the detailed review. We answer your questions below.
>
> > *“The model architecture seems very closely based on the HOPPITY baseline: the primary change seems to be to the multi-head graph encoder (which itself uses an existing GIN network for each head). While it’s exciting that this addition improves over the original, the resulting model therefore doesn’t seem to introduce a fundamentally new idea”*
>
> The multi-head graph encoder is one of the several contributing factors to the overall performance of Graphix: (1) the use of shorter and more meaningful edits, shown in Appendix B; (2) the grammar-guided decoding; (3) the multiple-head encoder and a new pre-training strategy. The following table illustrates these aspects, with improved results of Graphix-P when pre-trained with slightly larger sub-trees.
>
> | Model | Edit | Avg. small length | Avg. medium length | Model size | Small | Medium |
> | ------ | ----- | :-----: | :-----: | ----- | ----- | ----- |
> | HOPPITY | JSON diff edit | 14.1 | 35.2 | 1M | 7.30% | 1.21% |
> | HOPPITY | JSON diff edit + COPY | 7.3 | 12.6 | 1M | 8.47% | 2.91% |
> | GRAPHIX |Grammar-aware edit | 5.4 | 8.5 | 1M | 12.40% | 5.34% |
> | GRAPHIX |Grammar-aware edit | 5.4 | 8.5 | 32M | 19.92% | 9.56% |
>
> Overall, Graphix is a graph-based model that achieves near state-of-the-art results compared to large-scale Transformer models.
>
>
> > *“The authors heavily emphasize that the ‘smaller number of parameters’ of their model in comparison to the best-performing SOTA. This feels marginally interesting — and the ‘CodeT5-small’ model seems to perform very similarly to the model in this paper, with about the same order of magnitude of parameters. In fact, all of the best performing pre-trained models seem to perform about the same."*
>
> Thank you for pointing out the inaccuracy of our claim about the model size. This was a mistake! We have corrected the claim in the revision. In short, our model is about 2x smaller than CodeT5-small but performs better on the small data.
>
> > *“The evaluation omits a comparison to the closely related Yao 2021 model that is cited throughout the paper.”*
>
> The work by Yao et al., 2021 is related but applied to a different task of learning to edit given a change specification, as explained in https://openreview.net/forum?id=v9hAX77--cZ. Hoppity is the most direct comparison.
>
> > "Qualitative evaluation is intriguing but doesn’t yield clear conclusions. In particular, I’d be looking for evidence that ‘GRAPHIX demonstrates strong inductive biases in learning complex bug-fix patterns’ - or otherwise, some conclusion on the kinds of bugs that GRAPHIX tends to fix (largely syntactic or catchable by a linter? Requiring multiple edits in graph-space? Concentrated on specific kinds of operators?). One possibility is hand-coding the analyzed edits to categorize them and summarizing the results: without that, these examples are interesting but inconclusive."
>
> Indeed, we categorize correct and wrong fixes by Graphix and summarize the results in Appendix F and Appendix G. The anecdotal examples qualitatively show the kinds of bugs Graphix is able fo fix for the particular test benchmark. One advantage we want to emphasize with our approach is that examining the edits is much easier than comparing pairs of buggy and fixed code generated by translation-based models.
>
> > "The pre-training regime, which is presented as a contribution of this paper, does not seem to produce markedly stronger results."
>
> We agree that the pre-training does not produce stronger results. This is partly due to the scale of our pre-training experiment in terms of the sizes of corrupted sub-trees and the runtime cost of our model.
>
> As *Reviewer V4AB* and *Reviewer AZoW* suggested, we try with larger sub-trees from 3 to 8 nodes and continue training the existing pre-trained model. Note that we choose  8 nodes at maximum due to a scalability issue that we address in the response to *Reviewer AZoW*. We simplify the sequence of ADD operations with a mixed of ADD and COPY in the synthetic edits if there is a matched sub-tree found in other parts in the AST.
>
> We pre-train the model for 2 more epochs and fine-tune Graphix again. The results are shown in the following Table where we see some improvements on both small and medium datasets. The results suggest that pre-training with larger sub-trees (typically 15% corruption rate in pre-training Transformer models) may help further.
>
> | Pre-trained | Model	| Model Size	| Small	| Medium |
> | ----- | ----- | :-----: | ----- | ----- |
> | No | GRAPHIX | 32M	| 18.20%	| 9.19% |
> | Sub-tree of 2-6 nodes	| GRAPHIX-P	| 32M	| 19.81%	| 8.81% |
> | Sub-tree removal of 2-8 nodes + COPY op	| GRAPHIX-P	| 32M	| 19.92%	| 9.56% |
>
> Finally, we note the new results are on the abstract dataset. We will obtain similar results on the concrete code and include all in our revision.

---

> > ### Comment · Reviewer_z3GX · 2021-12-05
> > **Response to reviews**
> >
> > Thank you to the authors for their detailed response -- and apologies for the delay in response.
> >
> > As other reviewers have noted, the changes suggested by other reviewers -- and made by the authors -- are substantial, likely enough to require re-review of a revised paper.
> >
> > Further, I actually would maintain my primary statement from above: while the various stated contributions here may be important to the practical application of graph networks to this regime, they do not, in this reviewer's opinion, represent a substantially surprising or novel contribution beyond the initial application of graph networks to this problem.
> >
> > Therefore, I am keeping my score.

---

### Decision · Program_Chairs · 2022-01-20

**Decision:**

Reject

**Comment:**

This paper presents an approach for machine learning to fix programming errors via edits to abstract syntax trees. The main contributions are a pretraining scheme based on masking out subtrees and some minor architectural modifications compared to previous work. Reviewers found the paper to contain a significant amount of work, but there are some questions about significance relative to previous work that framed the problem similarly, and about experimental methodology. Authors did a great deal of work in the rebuttal to address many of the experimental methodology questions, but this also introduced substantial unreviewed changes to the model, the pretraining approach, and the experiments. In total, the remaining concerns about significance and the substantial changes lead us to recommend that this paper be revised and resubmitted to the next conference.